# Design considerations for a hierarchical semantic compositional framework for medical natural language understanding

**Ricky K. Taira**[1]*, **Anders O. Garlid**[1], **William Speier**[1,2]

**1** Medical and Imaging Informatics (MII) Group, Department of Radiological Sciences, University of California, Los Angeles, Los Angeles, California, United States of America, **2** Department of Bioengineering, University of California, Los Angeles, Los Angeles, California, United States of America

* rtaira@ucla.edu

## Abstract

Medical natural language processing (NLP) systems are a key enabling technology for transforming Big Data from clinical report repositories to information used to support disease models and validate intervention methods. However, current medical NLP systems fall considerably short when faced with the task of logically interpreting clinical text. In this paper, we describe a framework inspired by mechanisms of human cognition in an attempt to jump the NLP performance curve. The design centers on a hierarchical semantic compositional model (HSCM), which provides an internal substrate for guiding the interpretation process. The paper describes insights from four key cognitive aspects: semantic memory, semantic composition, semantic activation, and hierarchical predictive coding. We discuss the design of a generative semantic model and an associated semantic parser used to transform a free-text sentence into a logical representation of its meaning. The paper discusses supportive and antagonistic arguments for the key features of the architecture as a long-term foundational framework.

**Data Availability Statement:** This paper is a concept paper. There is no data associated with this paper.

## Introduction

Natural language processing (NLP) of clinical reports is an important area of research in medical informatics. It is considered a key enabling technology for transforming unstructured Big Data from clinical repositories into a computer-understandable representation that would allow for compiling phenotypic observations and treatments from a large number of patients [1, 2]. These curated structured databases can then potentially be used to build individually tailored predictive disease models and/or assist in identifying new patient stratification principles for targeted therapies [3–5]. A comprehensive review of the tasks and applications that involve NLP in the medical field are given in [6, 7].

Bibliographic reviews in the field of medical informatics have reported NLP-related research to rank among the most cited topics [8] with an increasing number of publications since at least 2007 [9]. Publicly available de-identified clinical data sets are now increasingly available for researchers. Community-wide standards for tagging and representation of NLP

**Funding:** This work was supported by funds from the National Institutes of Health grants R01-CA226079, R01-LM012309, R01-CA157533, R01-LM011333, and U24-AI117966. There was no additional external funding received for this study.

**Competing interests:** The authors have declared that no competing interests exist.

semantic constituents (e.g., concepts and relations) are being actively defined [10–15]. Cooperative publicly available toolkits and development environments are actively being contributed to and supported (*e.g.*, Open Health NLP Consortium [16]). New application areas continue to arise. Yet, despite these efforts and the long history of medical NLP as a focused area of research, the ability to perform deep understanding of clinical notes by computers remains elusive and generally far from the abilities of human cognition. The driving need for a deep understanding of medical text was emphasized as early as 2012 at a two-day workshop at the National Library of Medicine. At this meeting, prominent researchers in both general and biomedical NLP were invited to discuss directions and strategies that would lead to more efficient development of NLP solutions for diverse medical research applications [17]. These experts agreed that there is a need for a new paradigm involving the integration of statistics, linguistic knowledge, and domain knowledge. The late Dr. Donald Lindberg, then Director of the National Library of Medicine, emphasized the need for natural language understanding (NLU) over NLP. (For a description of the major issues related to the medical NLU problem, see the Discussion Section under the subheading "Comparing the Problem Circumstances of General versus Medical NLU").

The challenge of bringing a medical text understanding system closer to human capabilities is considerable. A key strategic design decision is specifying an overall system architecture to provide the framework for how various NLP tasks and knowledge sources interact. Reviews of architectural designs for medical NLP developments can be found in [6, 18, 19]. Currently, there are no agreed integrated models for deep understanding of clinical text.

Fig 1 shows an overview of the basic NLU mapping problem that transforms an input sequence of characters representing a sentence to a computer-understandable logical interpretation. Defining an ontological representation at this level depends upon the driving application, and in particular, the set of questions the NLU system should be able to answer. In other words, the fidelity of the "True Intended Meaning" can be formulated in terms of how well the

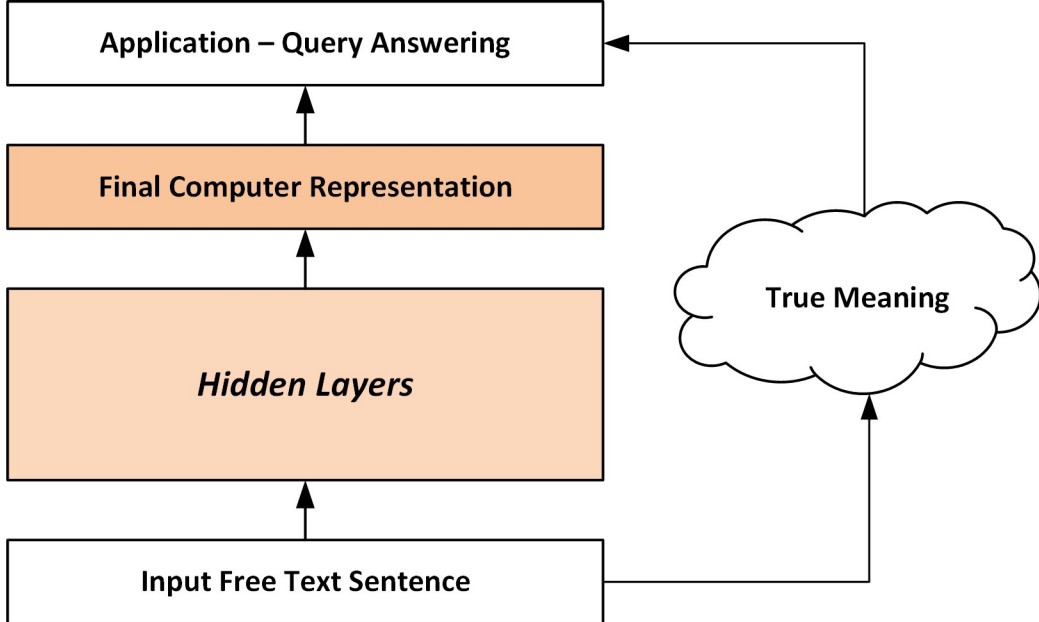

**Fig 1. Overall basic mapping problem.** The NLU problem maps the characters of a sentence to a conceptual representation of meaning. Defining the "internal semantic layers" is a key representational challenge.

NLU system can answer the set of application-driven queries in the paradigm of a Turing Test. A stricter account would also charge the system to explain how it derived its answers. For examples of various language level ambiguities within clinical text, see the Discussion Section under the subsection "Comparing the Problem Circumstances of General versus Medical NLU".

In order to address the large joint state space associated with this mapping, a number of internal layers are defined. Specifically, our approach involves factoring the NLU problem using a hierarchical semantic compositional model (HSCM). This structure enables a more efficient process of encoding sentence meaning by facilitating a generative model. The overall goal then is to navigate a sentence interpretation through the internal layers and states of the hierarchy using a predictive coding approach. Our design involves explicitly defining this structure in a way that parallels the manner in which humans compose meaning. This process contrasts with deep learning methods, which attempt to define these layers automatically based on training data and an objective function related to the specific query being addressed.

In this paper, we describe the conceptual design of a processing framework that can potentially serve as a foundational architecture for medical NLU applications. The architecture is designed to provide an overarching global semantic structure to organize a diversity of symbol-to-symbol mapping schemes in order to synthesize meaning from clinical text. These schemes can include a diversity of approaches, including rule-based, symbolic pattern matching, statistical inferencing methods, and deep learning approaches. The design decisions presented are in response to the known weakness inherent in data-driven approaches [20–22]. The main contribution of this paper is to present arguments for moving toward a symbolic-based NLU framework that is inspired by cognitive principles. We hope these arguments will stimulate much-needed open discussions toward the strategic developmental direction for building effective, sustainable clinical NLU solutions.

## Background

Although the exact nature of how humans can comprehend language so well and fast is still uncertain, there are four main inter-related ideas that are likely critical to our ability to comprehend language. (See, for example, [23, 24]). The four ideas are: 1) predefined abstract semantic representations; 2) semantic composition; 3) semantic activation; and 4) hierarchical predictive coding. A brief background describing how these ideas relate to natural language understanding is presented below.

### Predefined semantic representations

Humans possess what is known as semantic memory, which stores knowledge about declarative facts, ideas, meanings, concepts, and knowledge of the world we have acquired. Semantic memory is an integral part of human intelligence. Evidence for its physical existence is being investigated using fMRI activation studies, which show a continuous semantic space describing thousands of objects and action categories along the brain's cortical surface [25]. There is significant evidence that all animal brains have the ability to generalize and create categories and concepts and encode them in neurons, where each group of such cells is dedicated to a single category or concept [26]. Semantic memory can be viewed as precompiled informational structures primed for understanding language. When humans are presented with unfamiliar words or concepts, we adapt to our environment by evolving this representation (e.g., memory formation). Conversely, concepts no longer fitting for our "survival" may also cause semantic categories to be removed (memory loss). A radiologist, for example, might have a comprehensive abstract informational template for what is a tumoral mass compared to a non-medically

trained person. These templates allow language signals to be encoded efficiently into semantic memory, which in turn is primed for efficient interpretation. One important point related to semantic memory is that it is relatively comprehensive. It encompasses a representation for conceivably every discussion item and characterizes the capacity for which a person can understand language. Additionally, the stored summary representation used by the brain allows for the production and comprehension of sentences beyond those that have been experienced [27, 28]. With respect to medical NLU, capturing the meaning of a sentence requires the development of a sufficiently rich representation model suitable for its targeted situational use. Circumscribing the scope of sanctioned interpretations is part of the application domain modeling problem and, in general, there is a need to create application-specific (i.e., situational or "realism-based") ontologies and semantic models [29, 30].

## Semantic composition

Utilizing a compositional approach to meaning representation is an idea deeply rooted in language theory [31]. Composition allows a system to have a large descriptive capacity utilizing combinations of more elementary units. The rationale for the approach can be summarized as follows. Firstly, a direct mapping from text to sentence-level logical interpretation is unreasonable given the variability of free text and the large state space associated with the universe of all possible logical interpretations. To deal with these difficulties, it is typical to introduce layers of intermediate structure representing sub-interpretations [32, 33]. This composition significantly reduces the dimensionality of the mapping problem through independence assumptions. Secondly, humans understand text at several conceptual levels [34]. For example, humans can derive meaning from text at the morphological level (e.g., word endings), lexical level, within the context of a syntactic phrase, or within predicate argument constructions. Thirdly, cognitive science research generally views language as a generative process [35–37]. This implies that the language can produce an infinite number of sentences from its basic constructions as well as understand sentences it has never seen. From a cognitive point of view, an effective composition of a sentence implies that all the information that a human would expect to decipher from the sentence should be extractable from the compositional representation [38]. This structure implies that any questions that could be answered from the meaning of the sentence should be answerable from the representation alone (see Fig 1). That is, we do not lose any information stated within the sentence by factoring it into components that are themselves meaningful at various levels of semantic abstraction.

## Semantic activation networks

One powerful feature of the brain is the connectivity of its memory units. This connectivity allows the brain to support the notion of priming, in which memory units within the brain's semantic network are activated in such a way as to prepare the cognition system for encoding to-be-interpreted language signals [39]. That is, humans rely on a significant amount of knowledge in which words activate a cascade of semantically related concepts, relevant scenarios of their use, and models of reality [40, 41]. For example, mentioning a "tumor" within a medical report could prime an oncologist's brain to expect various clinical characteristics and referents of this topic concept. The mention of the phrase "located in" primes the cognitive system to expect a description of a spatial location. Connections among semantic memory units allow recalling related concepts to an activated concept, resulting in a functional integration of brain areas and the spreading of activated semantic units. Depending upon the types of entailment relationships coded within the network, the spreading of the activation can differ. This spreading activation builds a dynamic semantic field that primes the brain for maximizing

comprehension and introduces relevant context to elevate interpretation in response to and in anticipation of the given input [42]. The connectivity of semantic units is based largely on experience and knowledge. Knowledge aspects may involve a hierarchical typing system that humans commonly use to categorize objects, while experience aspects may be used to efficiently navigate to associated concepts based on past personal encounters [40]. Of note, the configuration of the network is highly fluid. Mounting evidence suggests that such network reconfiguration is necessary to help keep the overall cognitive system in healthy balance [43].

## Hierarchical predictive coding

Hierarchical predictive coding seems to be a fundamental mechanism for human cognition, involved in both vision [44] and language processing [45]. The brain uses it to solve seemingly intractable problems (e.g., scene analysis and language understanding) involving sensory inputs (e.g., visual or auditory signals) in a highly efficient manner. The central idea is that the brain is an organ of prediction guided by a hierarchical generative model of how it understands the world. Interpretation is seen as a process of minimizing free energy. Free energy is small when internal neural representations can accurately predict lower level inputs. Instead of minimizing the entropy of the interpretations, the strategy is to minimize the entropy of the observations (Free Energy Principle). Operationally, predictive coding refers to a processing paradigm that utilizes an adaptive strategy for hierarchically interpreting input sensory signals using a hybrid top-down and bottom-up processing approach. The top-down strategy uses lower level cues to posit upper level hypotheses (*i.e.*, causes) that are then tested based on evidence from lower level inputs (*i.e.*, observations). As part of the conceptual knowledge base of the brain, a predictive algorithm assesses the virtual situation that given an upper level hypothesis prior, what is its likelihood based on the state of the lower level inputs. Top-down processing will explain away (by predicting) only those elements of the driving signal that conform to (and hence are predicted by) the current winning hypothesis. The higher-level guesses are thus acting as priors for the lower-level processing in the fashion of so-called "empirical Bayes" [46] (such methods use their own internal target data sets to estimate the prior distribution: a kind of bootstrapping that exploits the statistical independencies that characterize hierarchical models). When a prediction is accepted, the system updates the higher-level priors to posteriors. The bottom-up processing relates to carrying lower level evidence that cannot be accounted for by the top-down predictions to higher levels as residual prediction errors. Thus, the better the top-down matches, the less we see prediction errors propagating up the hierarchy. Upper levels of processing provide greater context to interpret these residual errors (i.e., unaccounted tokens) due to the hypotheses that have been previously crystallized. Thus, in predictive coding, navigating the interpretation hierarchy relies on transitioning through the system's internal states by utilizing a cascade of top-down predictions to move up the hierarchy. The interesting aspect of the paradigm is the utilization of both successful predictions (as defined by some tolerable error rate) and unsuccessful predictions related to the input. This is an application of the idea of "analysis-by-synthesis" [47–50]. The processing paradigm also supports the notion that language comprehension is a form of abductive reasoning [51] in which the process of interpreting sentences in discourse can be viewed as the process of providing the best explanation of why the sentence would be true. In this processing model of the brain, hierarchical predictive coding [52] can be seen as a form of Bayesian filtering (least surprising interpretation) [44, 53–55].

## Methods

In this section, we first introduce the overall NLU problem highlighting the need for a predefined compositional structure. We then describe the elements and features of our central

HSCM knowledge base. The model design contains elements that emulate the cognitive features of semantic memory, semantic activation, and semantic composition. The design of the semantic parser, which executes on an input sentence to derive an ontologic representation of the meaning of the sentence, is then described. Finally, we highlight important principles employed by the architecture, which provide the basis for what we believe to be a solid foundation for future growth. Note that, for brevity, we focus only on foundational architectural issues (i.e., what should be done) and leave specific implementations (i.e., how it should be done) with respect to grammars, classifiers, and specific knowledge sources, as open options within the framework.

## Problem definition and overview

**Hierarchical semantic compositional model.** The HSCM knowledge source defines the "hidden layers" of the NLU mapping problem. Fig 2 shows an overview of the elements associated with the model. These elements can be described in terms of semantic constituents, semantic composition grammars, semantic networks, and a query processor. The HSCM addresses the need for an NLU system to possess a comprehensive internal representation for the universe of sentences it intends to understand (e.g., sentences that describe a tumor in a radiology report), which is the semantic substrate needed to encode meaning (see Fig 2, Box 1). The semantic constituents correspond to semantic memory elements within the cognitive paradigm. Defining the constituents within the model is an open research question and must be approached with caution since navigating and maintaining the knowledge source becomes increasingly difficult as the number of nodes in the hierarchy increases. It is thus imperative to apply various organizing principles including methods for building medical ontologies [56], methods for analyzing complex systems [57, 58], and methods involving problem decomposition [59] (e.g., abstraction, encapsulation, modularity, and inheritance). Referring to prior efforts in building semantic grammars and semantic frames for both medical and general NLP can also be productive [60, 61]. See, for example, work by the Linguistic Data Consortium and the Abstract Meaning Representation [62].

In practice, defining the semantic compositional model for a class of target sentences is not straightforward and can evolve to a variety of configurations. Accurately capturing the level of specificity required by the anticipated driving queries is an exercise in carefully decomposing each level of semantic detail. Topic-centric (e.g., tumoral mass) corpus-based (thoracic radiology reports) methods can be applied in general [63–65]. Alternatively, one could approach the problem from a syntactic point of view and proceed to learn, for example, the most common verbs and their related semantic constructions [66, 67]. As previously mentioned, the semantic nodes provide a template to encode language meaning at various levels of complexity. Although there have been efforts in the literature to define the semantic primitives and higher-order constituents for medical NLP, the specification of the constituent nodes is often by necessity a personal and situational effort. (Like the brain, we constantly update our internal expectation models of stimulus from our environment). Both bottom-up ("compositionality principle" [68]) and top-down ("context principle" [69]) methods for modeling semantic nodes are useful in designing appropriate levels of abstraction and organization. Fig 3 shows a rough schematic of this semantic generative representation. For diagrammatic purposes, we characterize various semantics constituents within domains according to specificity and/or semantic richness. A brief description of these broad node types is given below.

**Semantic constituents.** *Semantic layer 0—Surface words.* The hierarchical layering starts with a character stream for a given sentence. The first layer performs an initial surface level (i.e., the verbatim string) grouping of characters into words. A factory of tokenization schemes

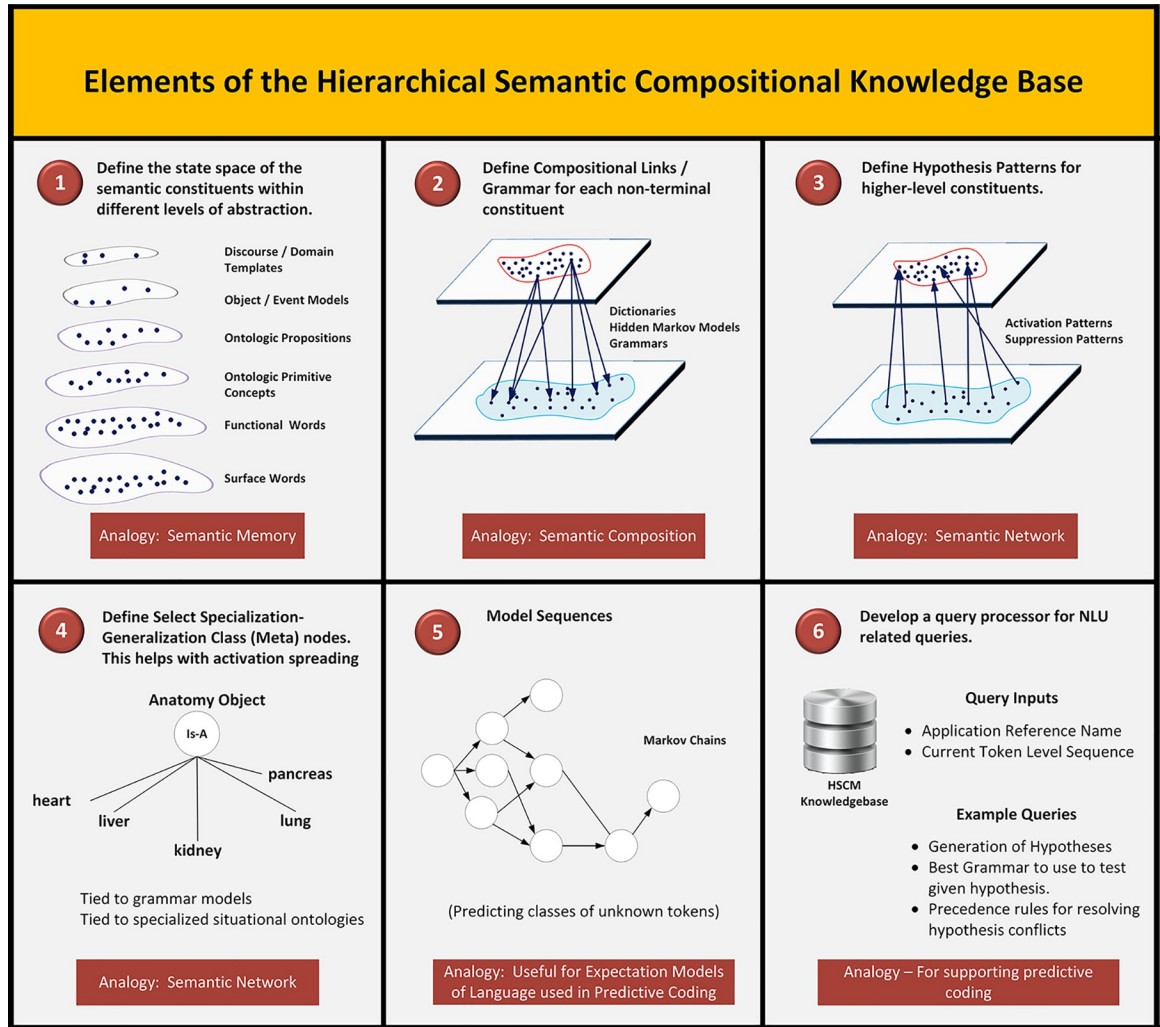

**Fig 2. Tasks associated with the construction of the hierarchical compositional semantic model.**

can be used to parse the character stream into a sequence of surface word tokens. Rule bases can be used to address dashes, slashes, apostrophes, and parentheticals [70].

*Semantic layer 1—Functional words*. Functional words can be defined as the primitive semantic constituents of a language. The functional definition of a word reflects how the system will strategize making semantic sense for a given segment of text. Different strategies for word-level tokenization will lead an NLP system to process a given input text in different ways. Mapping surface words to functional words involves a number of subproblems including: a) spelling corrections; b) identification of idiomatic expressions (e.g., throw up); d) identification of collocations (e.g., vena cava, computed tomography, and medical center); e) identification and/or parsing of symbol expressions; f) expansion and interpretation of abbreviations and acronyms; and e) decomposition of compound words. Commons knowledge sources used to address these subproblems include: idiomatic dictionaries, lexicons of common medical term collocations [71], and co-occurrence phrase chunking models utilizing simple *t*-tests [72].

*Semantic layer 2—Ontological primitives*. Ontologic primitives represent the lowest level internal HSCM constituents and are abstract units of meaning that are sanctioned by the interpretation system. Example concepts include nodes for numbers, property names, property

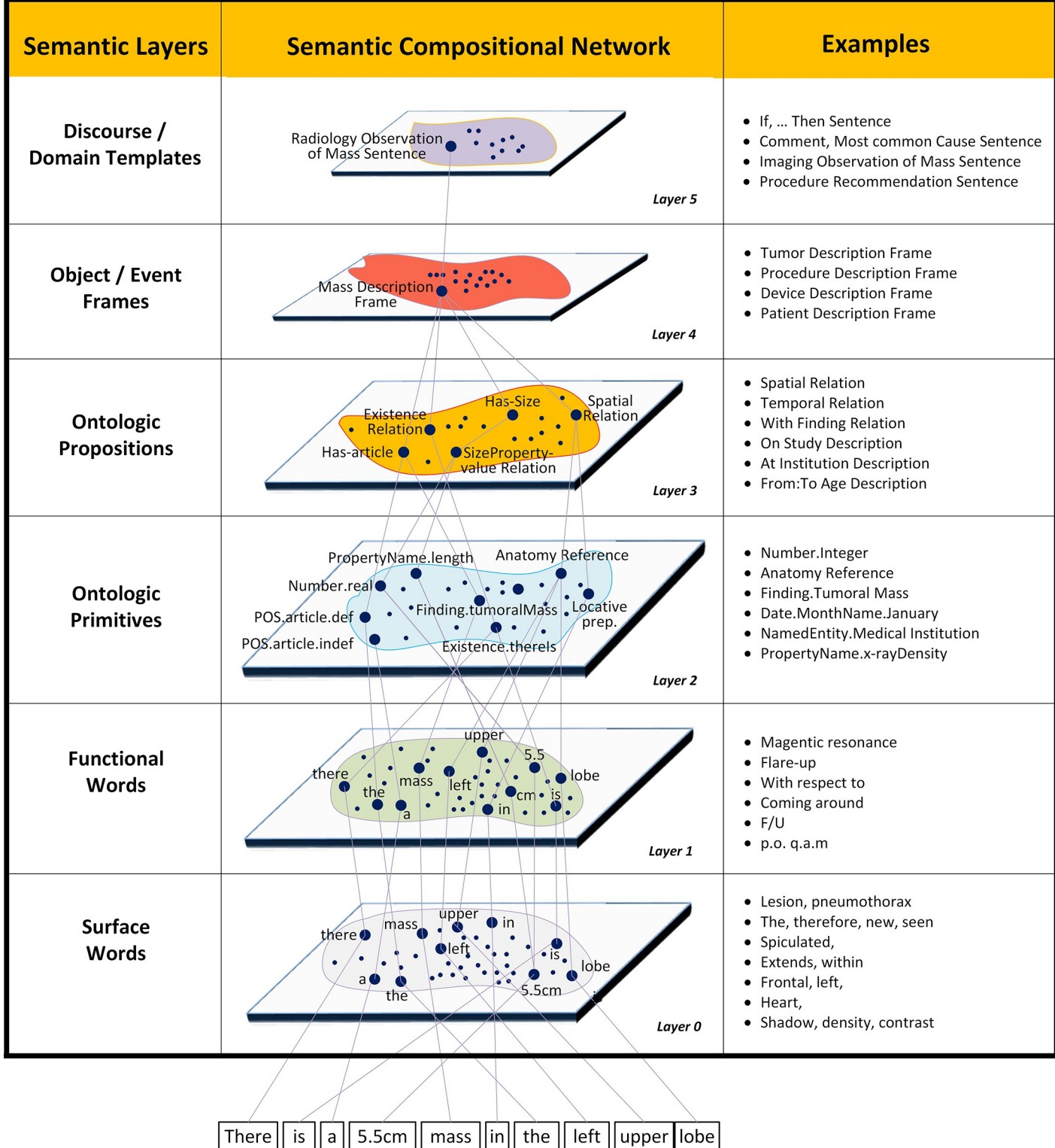

**Fig 3. Layers and example node instances for the HSCM.** An example sentence illustrates how the input tokens can be interpreted by instantiating network paths through the model. Each plane contains the domain of semantic constituents for the given abstraction level. For visual simplicity, arrows should be assumed to point downward to indicate compositionality.

values, certainty, medical procedure names, anatomy descriptions and medical conditions. Defining the granularity of these primitive constituents can be a challenging task often dictated by the particular application under consideration. For example, with size measurements, an application may simply want to parse the phrase "5cm x 4cm x 3cm" versus an alternative application which may require the internal semantics to be specified (i.e., the individual dimensions, units, and values). These choices in representation will influence strategies used for parsing (e.g., finite state machines or hidden Markov models). Some functional words, like "extends" and adverbs, are only identified within the HSCM in the context of higher order constituents such as other propositional constructions or predicate-argument structures due to their varying contextual roles across these targets. Thus, not all functional words will map directly to ontologic primitives.

*Semantic layer 3—Ontological propositions.* Moving up the semantic compositional hierarchy, lower level constituents continue to compose higher-level constructions. Propositions can be thought of as basic units of information (Finding Y is interpreted as Disease X). This level of semantic nodes include descriptions of properties and their values (*e.g.*, "size of 2.2cm x 3.0cm"), locative prepositions (*e.g.*, "within the right lower lobe of the lung"), temporal relations (*e.g.*, "within the last two weeks"), and various degrees of completion of predicate-argument structures (*e.g.*, "extends from the third to the fifth intercostal space"). Note that we can define complex propositions that provide more detailed descriptions by allowing the arguments of a proposition to be HSCM nodes at any level of abstraction. These arguments of propositions can include such node types such as ontologic primitives, other ontologic propositions, or higher-level object/event frames. For example, a spatial relation proposition could be formed using an anatomy frame for its location argument. A proposition describing an entity's size (e.g., "mass is 5cm in cranio-caudal dimension") could be constructed using a quantification relation frame and a size measurement frame.

*Semantic layer 4—Object / event frames.* These high-level nodes define comprehensive representational templates for targeted entities and events. Again, the definition of these node descriptions (i.e., their attributes) should be determined by formal ontology design and frame-based semantics methods. The richness of these nodes can be seen, for example, in defining a semantic entity frame for a tumoral mass. The specification of a mass includes a timeline of its states. A state is characterized by a collection of observations at a particular point in time. The observation description, in turn, is composed of a reference to a procedure and the various measurements associated with a property (e.g., size may be associated with three linear measurements). A procedure description is composed of a description of date, facilities, devices, and methodological protocols.

*Semantic layer 5—Discourse and domain-specific templates.* Conceptually, one could include even richer templates (e.g., application domains) at the sentence level and beyond, which comprehensively capture a more extensive range of semantic abstractions and text spans. Examples of such constructions include timelines, procedure-specific structured reports such as BiR-ADS, topic or procedure specific discourse templates, and phenomenon-centric disease models [73]). Discourse templates provide an expectation model between a speaker (e.g., specialist) and listener (e.g., referring physician). An example of a discourse model topic would be the expected communicative goals of a radiologist describing a patient's smoking habits in the context of determining whether a patient is eligible for lung cancer screening. In particular, this discourse model can be used to disambiguate instances of ellipsis commonly observed in this domain (e.g., incomplete specification of units of pack-years, which can be implicated from the discourse model). These high-level semantic templates are useful because they can be linked to application-specific queries. For example, an instance of a radiology mass template

could be used by a lung cancer screening application to help determine whether a patient might be eligible for a particular screening protocol.

**Semantic linkages.** *Downward links for compositionality.* As part of the HSCM, a number of semantic links are defined. Downward links define the constituents that can synthesize an upper level node (see Fig 2, Box 2). Each upper level composite node has its own methods for its grammatical construction. In practice, a variety of methods can perform these mappings. For example, the mappings may be implemented using dictionaries, lexico-semantic-syntactic patterns, finite state machines, context free grammars, or hidden sequence methods (e.g., Bayesian and deep learning methods). These mappings define the constituents, their sequencing, and the context for a valid construction. The methods of choice depend on the state space associated with their mapping, which in turn depends on the richness of the compositional model. As a note, although we use the term "downward link" to emphasize compositionality, in practice, the rules for construction are quite flexible so that the composition of higher order constituents (e.g., propositions, frames, and discourse templates) can be constructed using a variety of elements. For example, the arguments for a proposition can be filled with another ontologic proposition or even an ontologic frame. See the example described in the Parser Design section of this paper for further details and specific examples of these possibilities.

*Semantic activation network.* In addition to downward compositional links, we define links from lower level semantic types to higher-level semantic constituents (see Fig 2, Box 3). These upward links activate a process that identifies plausible hypotheses for constituents higher up in the HSCM given a set of tokens for a given sentence. For example, the word "extending" would trigger a hypothesis for instantiating the higher-level propositional template corresponding to the "extends" predicate argument structure. This link would then prime the system to activate the associated grammar to search for identifying modifiers and arguments associated with the semantic model for the "extends" proposition. Activation of hypotheses also occurs by exploring the ontologic attributes of entities that have been identified. For example, identifying the concept "tumoral mass" would automatically activate the grammars for all the attribute concepts associated with the ontologic definition of a tumoral mass (e.g., size, shape, radiographic density, and border architecture). These upward links can thus be seen as a means for allowing the parser to search for paths within the internal semantic hierarchical model to identify plausible interpretations of the input sentence. While downward compositional links are designed for high precision, the upward links that activate higher-level semantic hypotheses are designed to emphasize high recall. In contrast to upward activation links, suppression links can be activated by true negative language patterns to rule out a hypothesis. For example, the word 'mass' in the context of the phrase 'bone mass density" could use the lexical pattern "bone mass density" as a suppression pattern for the hypothesis of a (tumoral) mass concept.

The semantic activation network can also be extended using generalization-specialization links between HSCM nodes (see Fig 2, Box 4). For example, the general 'anatomy class' concept could include the subclass nodes "heart anatomy", "lung anatomy", and "liver anatomy". These concept relationships allow the HSCM to include such class level meta-nodes (i.e., anatomy class) that encapsulate the grammar model for the superclass. Thus, any subclass member (e.g., lung) can activate the hypothesis of the existence of an instance of the superclass. In processing a sentence, this implies that a subclass member (e.g., "lung") can activate the grammar attached to its superclass (e.g., anatomy concept).

**HSCM query processing.** Associated with the HSCM knowledge base is a query processing manager (see Fig 2, Box 6). It supports the following basic queries:

1. Retrieve all plausible hypotheses for the given token sequence and the given application profile. This task aims to identify patterns in the input token sequence that can activate upward links to higher-order semantic nodes within the HSCM. The query manager returns the set of plausible hypotheses, with each hypothesis corresponding to a node description in the HSCM. The hypothesis activation pattern can be influenced by the application such as when there exists a specific situational ontology tied to the given application.

2. Determine the most likely semantic node assignment for an unknown token within a sentence. For example, typing errors can be relatively common in medical reports. Various methods can be used to address this query including sequence language models (see Fig 2, Box 5), spelling correction algorithms, and context sensitive deep learning methods.

3. Given two incompatible hypotheses, decide which should be given precedence. The HSCM model maintains a decision model for resolving instances where two hypotheses within a sentence are conflicting due to overlapping tokens. Features such as compositional dependencies, token spans comparisons, and contextualized precedence ordering rules are maintained within the HSCM knowledge base.

4. Determine the semantic compatibility between two constituents. In cases in which the semantic parser (see next section) cannot combine a token into the overall semantic parse due to inadequacies within the compositional grammar, the query processor can ask the question: can the unattached token serve as an attribute for any of the other tokens within the sentence. For example, upon encountering an agrammatical sentence such as: "Mass, June 2020, 2.3cm in right lung, spiculated margins" if the concept "spiculated margins" is left unaccounted for by the grammar, the query processor can explore the HSCM concept "Mass", examining the property space contained as part of its logical representation. In effect, the query processor would convey to the client that the token "spiculated margins" is compatible with the real world logical understanding that it is a sanctioned feature for the concept of a "mass".

Examples for each of these query classes are provided in the following section describing the parser design.

**Parser design.**   Parsing a sentence involves transforming an input sequence of characters into well-formed logical representations sanctioned by the hierarchical compositional model. The semantic parser utilizes the main ideas from a hierarchical predictive coding paradigm and assumes the following HSCM knowledge sources are available: 1) a comprehensive internal model of the semantic constituents; 2) the associated grammar for synthesizing such constituents; and 3) hypothesis activation link definitions. Intuitively, the sentence input (e.g., tokens) is passed through a series of progressively finer-grained processing levels as described below. In this section, we use the running sentence example below to illustrate the overall processor steps.

[**Ex-1**] "*There is a 5.5cm mass in the left upper lobe.*"

Fig 3 shows the transformations through the HSCM for Ex-1. Fig 4 shows the transformation through various processing levels of refinement as executed by the parser. Note that the references to levels L0, L1, L2, L4, and L5 in the discussion of the parser (see Fig 4) are unrelated to the reference of the term "layer" in the HSCM representation.

**Preprocessing steps.**   The first two processing levels, L0 and L1, utilize standard NLP methods to handle relatively simple but useful, tasks and are briefly described below.

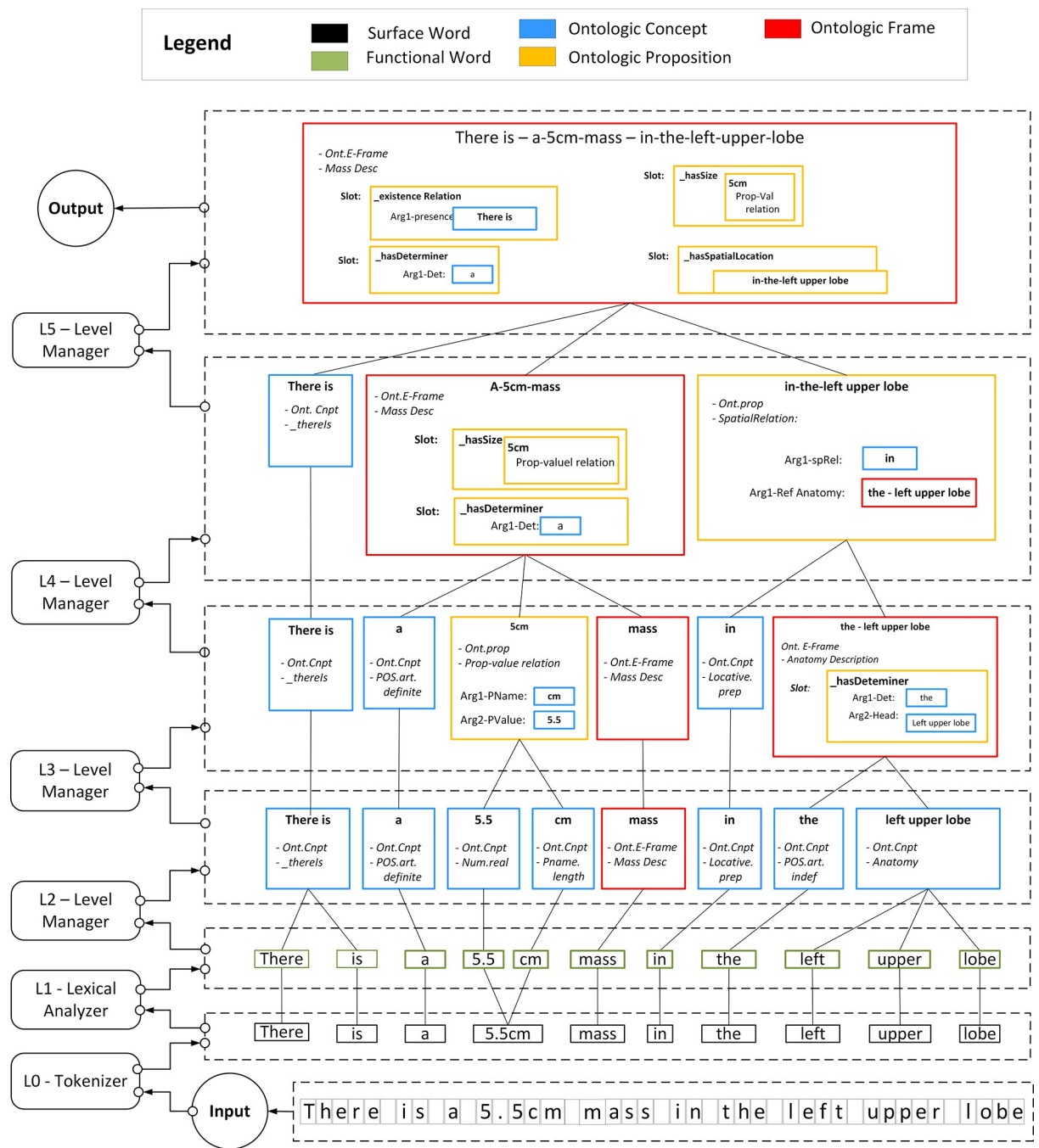

**Fig 4. Parser execution diagram for Ex-1.** The parser process involves iteratively transforming input tokens into higher levels of semantic abstraction. Box colors of tokens correspond to the class of the semantic constituent within the HSCM. (See figure legend for color assignments. Frame label definitions are as follows: pName.size–size property name; Ont.Cnpt–ontologic concept; Num.real–real number; PName–property name; pValue–property value; Ont.E-Frame–ontologic entity frame; POS.art.indef–part-of-speech description, definite article; Locative.prep–locative preposition).

*L0 Tokenizer.* The character sequence is mapped to a surface and functional word token sequence. This step is initially performed using common delimiters for orthographic tokenization (e.g., whitespaces and brackets). Attention to the particular input character representation

**Table 1. Example of functional word features as assigned to sentence Ex-1 during the lexical analysis step.**

| Functional Word | L1 Semantic Class | POS |
|---|---|---|
| There is | relation.exist.be | connective |
| a | pos.indef_art | det |
| 5.5 | number | adjective |
| cm | propertyName.length | noun |
| mass | physobj.finding.abnormal | noun.sing |
| in | pos.in | preposition |
| the | pos.defin_art | determiner |
| left | propertyValue.spatial.direction | adjective |
| upper | propertyValue.spatial.direction | adjective |
| lobe | physobj.anatomy | noun.sing |

scheme is vital for the proper application of tokenization rules (e.g., ASCII, UTF-32, and EBC-DIC). Hyphens and slashes can be disambiguated using context sensitive pattern-based rules. Certain characters such as quotes may be completely ignored in the tokenization process. In our example, the L0 tokenization step results in ten surface word tokens as shown in Fig 4.

*L1 Lexical analyzer.* The L1 lexical analyzer processing task performs additional mappings of surface words to functional words and computes some initial word level features for each token. In identifying functional words, the lexical analyzer makes use of precompiled lexicons for drug names, abbreviations, and medical idioms. Example word level features computed by the lexical analyzer include morphological features of words, embedding assignments, context free semantic classes from a general medical dictionary, part-of-speech tags, and dependency syntactic parser linkages to other tokens in the sentence. Private to the lexical analyzer are domain-specific pre-compiled lexicons (e.g., radiology terms, drug names, and special symbols) that are used to assign an initial general word-level semantic class for each token. See Table 1 for example semantic class and part-of-speech features for Ex-1. The granularity of the L1 semantic tagset is similar to that of the UMLS semantic network. L1 semantics use an outline label-naming syntax to indicate class/subclass relationships (e.g., "physobj.anatomy". Note that the L1 semantics described here are not part of the HSCM model per se, but are used as features to facilitate upward HSCM mappings, as in the task of semantic activation. This mapping is primarily used as the starting point (i.e., prior) for generalizing word-level context for upper level interpretations, rather than for mapping to precise end-user meaning. Out-of-vocabulary terms are initially assigned an L1 semantic tag of _UNKNOWN. The HSCM query processor (see Fig 2, Box 6) can be consulted to posit initial labels as described above using predictive sequence models. The lexical analyzer also maintains a rule base containing hand-crafted sequence patterns to resolve some word sense ambiguities. These ambiguities can often be handled better at higher processing stage levels due to improved surrounding context. As a final note, observe in Fig 4 that the surface token "5.5cm" was parsed into the two functional word tokens of "5.5" and "cm". This particular parse is performed to ultimately extract the internal semantics of value and units of the length measurement separately. Further details of these first two processing levels can be found in [71, 74].

L1 semantic class and part-of-speech features assigned to the function word tokens for Ex-1. Note that the L1 semantic word classes are not part of the HSCM model and are used only as features of the functional word class. The semantics categories are adapted from [71].

**Predictive coding.** Starting from the third processing level (L2), the parser proceeds using the general ideas of hierarchical predictive coding. The parser performs an iterative procedure summarized as follows:

*a. Instantiate a level manager.* A level manager is instantiated to coordinate the global processing for the current stream of tokens (see Fig 4). The level manager can access the current stream of tokens and global knowledge of the report context and/or driving application (e.g., section heading, procedure description, and NLU task definition).

*b. Perform hypothesis generation process.* (see Fig 5, Box 1): The level manager queries the HSCM knowledge base to identify all possible HSCM hypotheses given the current set of tokens and situational context. The activation network is consulted for this task. The task is treated as a node retrieval problem for most likely HSCM constituents given the current tokens and driving application information. In our example, various functional word patterns can activate a hypothesis. The function word "lobe", for example, has an L1 semantic tag of "physobj.anatomy" which will activate the hypothesis of the HSCM ontologic concept "anatomy concept". Depending upon the driving application, a word such as "lobe" could activate a more specialized ontologic concept class. For example, for a hepatology application, the word 'lobe' could activate a specialized HSCM concept node "anatomy.liver". The specialized node can either inherit the grammar from the more general class or can include its own local grammar model in order to parse specific anatomic elements of the liver. Some tokens, like the word "mass", can activate hypotheses through several layers of the HSCM model. In Fig 3, the L2 level manager will note that the function word "mass" has an upward link to the ontologic

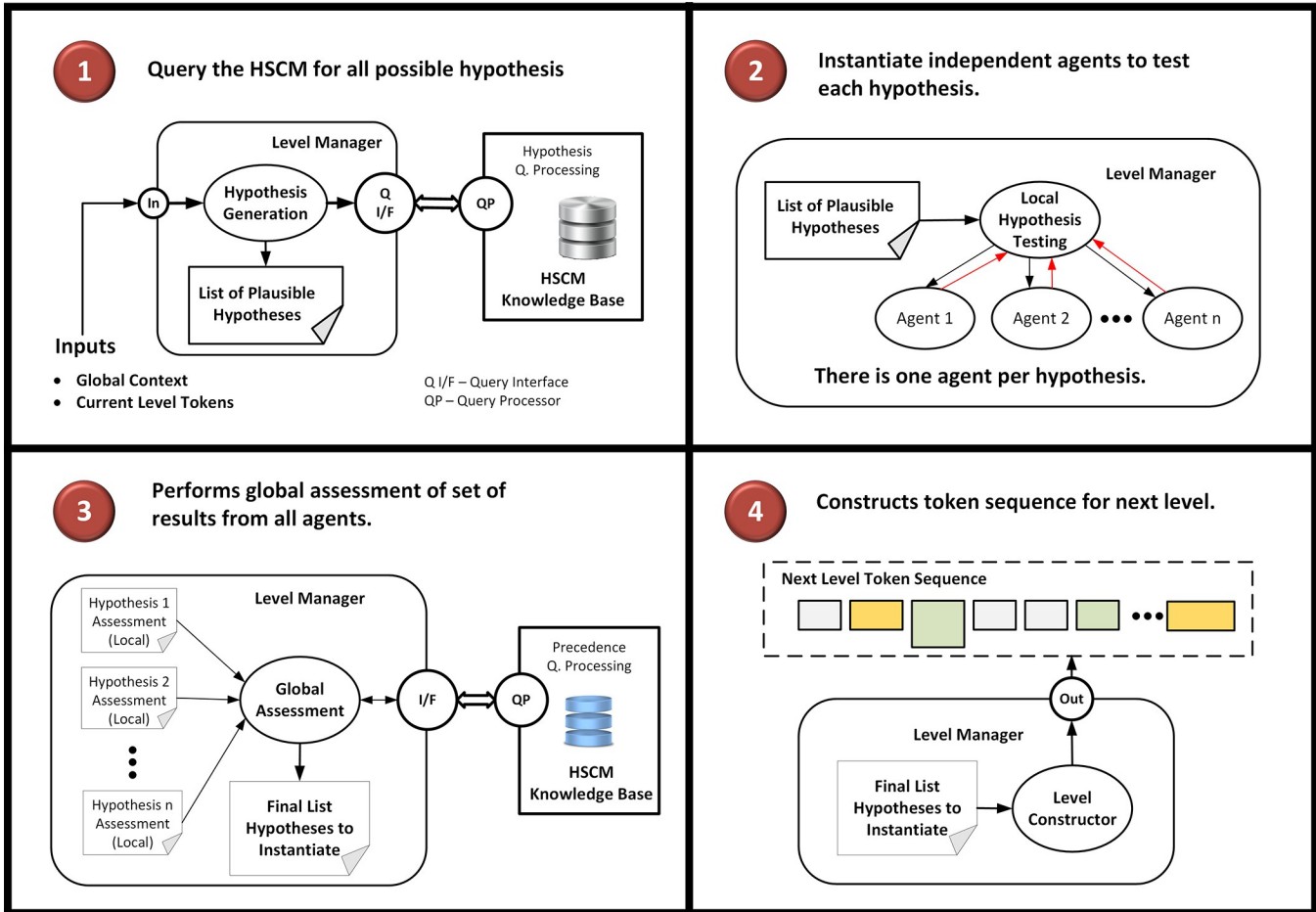

**Fig 5. Internal processes initiated by the level manager within the parser execution process.**

concept "Finding.tumoralMass" which in turn has an upward link to the HSCM model for the "Mass Description Frame". Interestingly, the "Mass Description Frame" can activates hypothesis pointing to the ontologic attributes associated with a tumoral mass. This causes a cascade of new hypotheses that includes each of the possible properties associated with the mass. For example, there is an HSCM node for "x-ray density signal intensity" associated with the radiological attributes of a mass. This hypothesis could then be used to characterize the phrase "low density" in the term "low density mass".

*c. Perform hypothesis testing process*. (see Fig 5, Box 2): The level manager activates a bank of agents to test each of the independent activated HSCM hypotheses utilizing their corresponding local grammars. Each higher-level hypothesis is then tested against the current level tokens to assess the validity of the hypothesis. An independent software agent is dispatched per hypothesis. Encapsulated within the semantic node being tested is a grammar model for its synthesis. Short spanning concepts (e.g., single word concepts such as "mass", "spiculated", and "well-circumscribed") can be identified with simple lexico-syntactic-semantic patterns that may include left and right local context. Longer, more complex instances, can be tested, using for example, a finite state machine grammar. Each hypothesis testing agent returns to the level manager a report regarding the truth of the hypothesis. If the hypothesis is true, the agent returns to the level manager the instance (or instances) of the hypothesized HSCM node. Note that the level manager can control which hypothesis testing algorithms (i.e., grammars) to apply depending upon the task application and/or prior failures under similar token context during lower levels of processing.

*d. Perform global level assessment of hypothesis testing results*. (see Fig 5, Box 3): The level manager receives all the results from each of the individual hypothesis testing agents. Again, note that each hypothesis is tested in isolation from all others, thus the need for a global consistency check. The level manager is responsible for adjudicating competitive and/or conflicting hypotheses in order to decide which, if any, should be ultimately instantiated. That is, it must decide which set of hypotheses can credibly explain the input sequence of tokens. Conflicts may arise due to overlapping token sequences. If there are no partial overlapping tokens, a simple rule to prefer the longer text span can be applied. For example, in Ex-1, the anatomy phrase hypothesis "left upper lobe" would have preference over the hypotheses for the individual tokens "left" (as an anatomic direction), "upper" (also as an anatomic direct), and "lobe". Ideally, two different hypotheses with the same text span should not occur in the HSCM model, and would be logged as an inconsistency in the model to be resolved by an adjudicating process. Imposing semantic constraints can can be applied to resolve syntactic attachment ambiguities and/or situations in which two hypotheses have partially overlapping token spans. A rule base or classifier can be consulted as part of the HSCM query capabilities (see Fig 5, box 3). For example, consider the sentence:

[**Ex-2**]: "There is mass in the right lower lobe that is still growing".

Two possible competing hypotheses for the token sequence "that is still growing" are the synthesis of an "Anatomy-Perturbation Frame" ("right lower lobe is still growing), or a "Mass-Finding Frame" ("mass is still growing").

Here, the HSCM manager would need to check whether the anatomy phrase "right lower lobe" is semantically in the role of an anatomic reference location or the subject of an anatomic description. From the valid construction of the spatial location predicate "in the right lower lobe", the HSCM query manager infers that the anatomy phrase is a reference location and thus can rule out its participation within the "Anatomy-Perturbation Frame." Thus, in general,

various types of complex dependency relationships and their respective ordering precedence can be maintained by the HSCM to resolve such conflicts.

*e, f. Instantiate next level token sequence.* (see Fig 5, Box 4): The last task of the level manager is to define the token sequence for the next iteration of processing. In Fig 4, the L2-Level manager has identified several 1-to-1 token mappings from the functional word level to elementary ontologic concepts for sentence Ex-1. For example, the surface words "5.5" and "cm" are mapped to the elementary concepts "number.real" and "property.length.unit". Note that the word "mass" was mapped to the high-level HSCM node referring to the "Mass Description Frame". The L2 manager combined the three tokens "left," "upper," and "lobe" into the general class of anatomy concept. Note that there is a reduction from 11 to 8 tokens as the parser progressed from level 2 to level 3 processing stages. Also, note that at higher processing levels of the example in Fig 4, the composition of instantiated tokens can be synthesized using a diversity of node types. For example, the L3 level manager synthesizes two ontologic concept nodes (the "5.5" and "cm" tokens) into a single ontologic proposition node ("Property-Value relation"). The L4 level manager constructs an ontologic proposition node describing a spatial relation from an ontologic concept node (viz., the "locative.preposition" concept "in") with an ontologic entity frame (viz., "Anatomy Description" frame "the-left upper lobe"). As a final example, the L5 level manager defines the top level "Mass Description Frame" from an ontologic concept (viz., the "_thereIs" concept), an ontologic entity frame (viz., "Mass Description" Frame), and an ontologic preposition (viz., "Spatial Relation" preposition). Note also the application of a recursion grammar for the "Mass Description Frame." Finally, in defining the next level of tokens, any tokens that cannot be integrated or refined are simply percolated up to the next level processing stage. The idea is that these residual tokens will have a better chance to be interpreted by the HSCM at the next level, where the context for its interpretative role is stronger due to the reduced number of tokens and richer semantic elements.

**Features of the design.**   In this section, we discuss some of the notable features of the design and its rationale.

**Structure first.**   The most notable element of the design is the presence of the HSCM. The design emphasizes the need to impose a pre-defined internal structure governing the sentence interpretation process. This structure allows the system to factor the understanding task into a number of lower dimensionality problems. Without such structure, it is unlikely that a large clinical corpus alone could model all the contextual variability required for deep understanding. Assuming that the process of semantic compositionality accurately mirrors how humans would factor the interpretation for a given utterance, the structure of the semantic hierarchy will tend to be stable over time, although the stochastic nature of the network will vary across document domains [75]. Once the representation for interpreting a sentence can be established, then the process of acquiring the knowledge for how to navigate through the hierarchy becomes systematically clear. Each node maintains a local grammar model for how it is synthesized from its parts and context. The predefined semantic structure also greatly increases the probability of generating only plausible interpretations. For example, given a comprehensive frame model for a tumoral mass, the parser is guided by the semantic selectional constraints defined by the frame definition and thus can instantiate only property states of the mass that are sanctioned by the model.

**Multi-scale representation.**   The parser borrows ideas related to scale-space representation in which the input token sequence of words is iteratively transformed to coarser levels of representation. Each level results in either a reduction or semantic refinement of the tokens from the previous level. Higher-level semantic abstractions summarize structures at finer scales in a manner controlled by their defining semantic grammar. The multi-scale representation aim is to simplify further processing by compacting local details from the current level

token sequence. From a signal processing point of view, the constituents at coarser scales constitute simplifications of corresponding constituents at finer scales, a form of noise reduction [76]. The suppression of fine-scale details generally improves the surrounding context for making compositional aggregation decisions at higher processing levels.

**Pattern activation and recognition.** The brain primes itself to receive expected information by activating various semantic memory units. This activation allows the brain to bring into working memory prior semantic expectations for anticipated language signals. These activated nodes serve as hypotheses to be tested using "environmental sensors", which in our design are the semantic grammars associated with each HSCM node. Here the "environment" refers to the current sequence of tokens being analyzed by the parser within the context of the application. The activation in our design can be triggered in several ways:

1. Anchored triggering–where a detected base pattern activates associated grammar patterns for an HSCM constituent; For example, the string "cm" might activate the HSCM node for size which then activates the grammar for parsing a size expression (e.g., 4cm x 3cm).

2. Floating triggers—where grammar patterns are activated in any context. For example, we automatically activate existence phrase grammars for all medical sentences.

3. Cascading activation–where a low-level pattern can activate a higher-level semantic frame, which can activate patterns associated with frame's attributes and/or entailment relation relatives. For example, the string "mass" can activate the HSCM node for tumoral mass, which then triggers all the property attributes associated with a tumoral mass.

These adaptive activation strategies provide an efficient mechanism for realizing sensible hypotheses for an input sentence associated with plausible HSCM constituents. The activation process also improves global situational awareness of expected information. The framework thus provides the flexibility for integrating a variety of context-sensitive activation schemes that can include features such as application goals, document type, semantic results from prior document sentences, and external medical ontologies [77].

**Agent based architecture.** The conceptual design borrows ideas from distributed agents that act independently. At each level of parser processing, independent software agents are assigned to execute the testing of triggered HSCM hypotheses. Each processing level has a manager that administers these spawned agents. The level manager collects the evidence acquired by each agent to make a global set of actions for the current level of processing. The level manager makes decisions regarding competing hypotheses as well as decides which particular methods for a constituent should be activated. For example, in our work, we have a general semantic grammar for anatomy as well as a specialized more detailed grammar for eye anatomy [78]. The level manager provides the framework to incorporate multiple strategies for explaining away the input level tokens. This framework offers a flexible mechanism for integrating multiple approaches to solve identical problems (e.g., pattern based, probabilistic Markov models, finite state machines, etc.). This global knowledge of available methods and their strengths and weaknesses allows the system to identify the best algorithm for the current level environment and/or apply secondary, more generalized methods in the event that the current methods do not work satisfactorily. For example, ideas of topic centering [79] could be used to interpret residual tokens which are not satisfactorily accommodated by the HSCM grammar.

**Frame-based representation.** Level 2 and higher processing steps implement the key cognitive concept of a semantic frame [80–82]. Ontologic frames for medical entities are key representational candidates for structuring clinical phenotypes. Anchoring the representation around semantic frames allows the system to take advantage of key ideas such as object-

oriented descriptions, recursion, and procedural triggers [83]. The semantic frame representation is used both by the HSCM knowledge base and for characterizing token instances during parser execution.

**Predictive coding.**   The predictive coding feature of the parser utilizes a hybrid top-down and bottom-up approach to navigating a sentence through the HSCM. The top-down processing attempts to estimate a forward probability, $P(Evidence|Hypothesis)$, the likelihood, of a given activated hypothesis, which is generally easier to estimate than the inverse probability, $P(Hypothesis|Evidence)$, the posterior. For example, given a concept we wish to articulate, defining a local grammar is easier than testing a sequence of words and testing for every possible HSCM hypothesis. The top-level ("more cognitive") nodes in the HSCM intuitively correspond to increasingly abstract conceptualizations of the world, and these tend to capture or depend upon regularities that span larger text excerpts. The fact that there are many more arrows lower in the hierarchy indicates that the forward problem is generally easier than the reverse problem. The bottom up problem focuses mainly on estimating $P(Hypothesis)$ priors for HSCM nodes. The semantic activation step reflects an intelligent assignment of priors. The top-down step then focuses on using the semantic grammar for the hypothesis class, to test whether the evidence (i.e., current token sequence) can successively generate the hypothesis. Thus, the predictive coding step inverts the conventional view of bottom-up NLP processing. A successful hypothesis, i.e., one that can be explained away the observed tokens, then allows the priors to be updated to posteriors at the next iteration of processing in conformity with Bayes' theorem.

**Generative approach.**   The generative approach uses the predictive coding strategy to pull itself up the HSCM semantic interpretation hierarchy. The semantic structures at each level provide transparency for explaining how high-level interpretations are derived. This structure makes the framework relatively straightforward to debug. The generative approach provides a path for loosely-connected group efforts to develop a progressively capable system for an expanding scope of topics. Each group could develop shallow grammar models for relevant nodes. As the HSCM model matures and its representation becomes more stable, global optimization methods (e.g., various statistical / neural network models) can be applied.

## Discussion

The challenge of bringing a medical text understanding system closer to human capabilities is considerable. In this paper, we discuss a framework which we believe can serve as a foundational architecture for deep understanding of clinical text for diverse clinical problems. The presented framework is primarily knowledge-driven and currently heavily dependent upon manipulating symbolic representations. This framework contrasts the current trend of high performance NLP systems based on data-driven deep learning methods. Below, we present arguments for specific discussion items likely to be of concern regarding our strategic design.

### Comparing the Problem Circumstances of General versus Medical NLU

The first question one might ask is whether particular issues regarding the medical NLU problem that warrant moving towards a cognitive framework. Six perspective differences are presented below.

**Task-oriented.**   With respect to problem definition, the medical NLU system's value hinges exclusively on providing the necessary information to accomplish an "actionable" task [84]. Medical NLU systems thus are not intended to be general broad coverage applications, but instead targeted agents that are tasked to understand text at sufficient levels of detail and content to correctly guide a clinical or research action. It is important to realize that these tasks

can be high-stake and/or mission critical responsibilities, compared to an NLP system that may be searching for information stored on the web or in journal articles. Thus, ignoring tail distribution cases may be unacceptable. For example, suppose the NLU system is tasked to identify patients who should be screened for lung cancer, based on clinical reports describing their chest x-ray findings and smoking habits. Failure to identify such patients should not be hinged on idiosyncratic language including various difficult language phenomena (e.g., ellipsis, coreference resolution, presuppositional inferencing, and linguistic paraphrasing) used within these medical reports. As Dunietz points out, "the field of natural language processing is chasing the wrong goal" [85, 86]. This message implies that a robust NLU application must consider all possible relevant content in all possible contexts for the driving task. He likens current NLP research as analogous to "trying to become a professional sprinter by glancing around the gym and adopting any exercises that look hard." The paper also emphasizes the importance of staying task-based as opposed to method-based in order to address all possible comprehension intent issues.

**Limited task scope.** Clinical NLU problems focus on tasks with comparatively limited scope of understanding compared to general applications such as a robotics personal assistant application. There have been a number of potential target NLU applications identified across various medical domains [87]. Some examples include: 1) characterization of patient lifestyle habits (e.g., smoking, exercise, diet, alcohol consumption, use of recreational drugs) [88, 89]; 2) characterization of mental health conditions (e.g., depression, suicidal, psychiatric syndromes) [90]; 3) characterization of specific clinical findings (e.g., tumoral masses, aneurysms) [91]; 4) identification of patients matching disease screening protocols (e.g., lung cancer) [92]; and 5) characterization of interventions (e.g., tube placement, medications, surgical details) [12, 93]. Although the scope of the tasks can be seen as relatively narrow, the performance requirements concerning semantic granularity, explanatory competence, and error frequency and types can be demanding.

**Limited text pool.** The pool of possible sentences to be encountered by any single NLU application is relatively narrow. This limits the vocabulary size and complexity of the grammar compared to general free text. Clinical text, however, can be quite varied with respect to its formality, style, and flow, especially across medical domains. For example, radiology reports typically have a formal declarative language style with complete sentences. Primary physician notes often show abbreviated styles with sentence fragments. Discharge summaries often include lengthy sentences of episodic descriptions in the context of event timelines. An admission note can include a patient's own narrative description of their problem using lay language and/or foreign words. The content contained in a given type of report also can vary with respect to the experience of the authoring physician. For example, the reporting style of a novice physician (e.g., resident) can be significantly different from an experience physician (e.g., a novice might generate longer wordier reports that include extraneous descriptions of low-level findings) [94]. The above described variabilities can create challenges for an NLU system to infer the main points of a report communication due to the presence of challenging language aspects such as coordination, coreference resolution and ellipsis.

**Predefined language representation.** The output representation associated with a medical NLU task is predefined and highly structured. There are two main layers of representation: 1) a knowledge representation model associated with the written text per se; and 2) an ontologic based representation focused on the NLU task [95]. The first level builds a semantic representation from the input sentence-level perspective [96]. General knowledge representations for language provide the framework for semantic analysis and an effective structure for constraining the synthesis of semantic constituents [97]. Constituents at this level include word senses, predicate argument structures, and semantic frame definitions. The second level, the

application's ontologic representation, is intended to directly support queries related to the driving NLU task. This is related to the fact that the output representation is intended to directly support inferencing operations for the clinical decision task in question. For example, well-accepted de-facto reporting models such as BiRADS are specifically designed to infer appropriate patient management strategies [98]. The RECIST model is designed explicitly to standardize the reporting of cancer tumor characteristics in response to treatment [99]. The United States Preventive Services Task Force has defined an information model for patient smoking habits that can be used to infer candidate patients for lung cancer screening procedures [100].

**Speaker-listener model.** There is an underlying speaker-listening coupling that facilitates physician-to-physician communication [101]. That is, the author of a report has in mind the needs of the reader, and the reader has in mind the intentions of the author [102]. This is especially strong among individual physicians who routinely communicate findings and recommendations for given types of patient studies. Societal clinical reporting guidelines can synchronize the expected information content to be communicated between physicians for given types of investigations [103]. This allows the reader of a report to derive interpretations beyond what can be inferred from words alone by leveraging diverse pragmatic knowledge [104]. Thus, although various language complexities are common in medical reports (e.g., lexical and referential ambiguity, ellipsis, and punctuation ambiguity), they can often be mentally corrected from within this overarching expectation model [105]. Contrarily, without the assumed background knowledge, clinicians may be unable to impute the intended meaning of ambiguously written text ("Curse of knowledge") [106]. This can lead to misinterpretation of medical data and, therefore negative patient outcomes [107–109]. Some examples of various language understanding complexities observed in medical reports include:

- Underspecification of terms–e.g., the phrase "*left apex*" in a chest x-ray report refers to the anatomic term "*apex of the left upper lobe of the lung.*"

- Wrong use of valid terminology–Physicians may misuse or mis-interpret the meaning of obscure units of measure. For example, consider the sentence, "*The patient has a cumulative 30 year smoking history of 20 packs per year.*" The authoring physician, in this case, has misused the units of "packs per year". Since the sentence is a description of cumulative smoking history, the correct unit that should be stated is "pack-years." The two units, although sounding very similar, have very different meanings, with "pack per year" describing the rate of smoking behavior while "pack-years" is an integrated cumulative value for smoking history. The correct interpretation is essential, since the critical description of "pack-years" is used to determine eligibility for lung cancer screening.

- Punctuation Mis-use–In a medical report, physicians may ignore adding full-stop punctuation between sentences. The text is thus seen with multiple sentences running together. The reader's basic knowledge of syntax allows sentences to be mentally identified.

- Ambiguous pronoun references–Pronoun reference resolution often depends on the reader having knowledge regarding the physical state and/or dynamics of the referring object. Disambiguation depends on the reader's ability to mentally assess real world consistencies and/or test counterfactual hypotheses in order that the formulating interpretations of the text adhere to an expected model of the world [110].

- Temporal Ambiguity–For example, consider the sentence: "*Patient is widowed since 1972, no tobacco, no alcohol, lives alone, smoked 3 packs per day x 17 years,*" (taken from I2B2 smoking corpus) [111]. Here, the reader should assume that the patient currently does not smoke but

previously did smoke for 17 years, with the start time and end time of smoking history unspecified.

- Inferred information from practice guidelines–From a description of a medical conclusion, one can positively infer other useful information. For example, in the sentence: "*This patient satisfies age and smoking criteria for routine annual screening.*" This implies that, according to the 2018 American Cancer Society guidelines, that the patient is between 55 and 74 years of age, has a smoking history of at least 30 pack years, and either currently smokes or has quit within the past 15 years [112]. Note that this guideline can change and recently (2021) this guideline is being reviewed for revisions based on the latest scientific evidence.

This listener-speaker assumption allows the reporting physician to avoid excessive verbiage. The report need not make explicit every level of detail. In some cases, however, physicians can be overly detailed in a negative way. For example, pathologists have been shown to emphasize the completeness of details, but have largely ignored the ability of clinicians to comprehend such detail [113]. This can hinder the intent of the pathologist to convey a more detailed description about the nature of a patient's disease state, which could be useful for determining the best management strategies.

**Comprehensive evaluation required by medical NLU systems.** The evaluation procedure for medical NLU applications must go far beyond the technical assessments reported in general NLP studies. This is because the relative importance of various outcome measures is different within an applied field compared to a more basic computing field such as computer science. At the heart of the manner is the fact that in an applied field, development is application specific. By contrast, in a computing field, it is data centric. Thus, in a data-driven field, contributions are targeted toward how much data can be accounted for by the model, as well as the number of applications that can be supported. In contrast, medical NLU applications are evaluated with respect to their effect on clinical care [114]. The general data-driven fields maintains leaderboards for broad tasks, which are scored based on contingency statistics (e.g., precision, recall, AUROC). Performance is evaluated based on models developed on shared pre-defined training and test data. Strategies for handling difficult test cases are rarely reported. Medical NLU applications, however, require not only a technical evaluation component but also are subjected continuously to various levels of scrutiny over the lifetime of its deployment [115–117]. The evaluation is end-user focused in the sense of what the actual impact the application has on clinical care. Application-centric metrics can take on addressing questions such as the following:

- How many patient cases was the NLU system used?

- How much time and manpower did the system save?

- How much more time was required by the user to review a patient's record?

- How many times did the system agree with the expert within the context of actual clinical care?

- How many unexpected results lead to negative patient outcomes?

- How many times did the system improve patient outcomes?

- How many times was the system unavailable to provide a satisfactory answer? This might involve the inability of the system to provide a reasonable explanation.

- How responsive is the development team to correcting reported errors?

- How confident are clinicians in using the application?

The medical NLU system must be continuously evaluated with respect to its failures and how these failures are addressed. These failures must be evaluated not only from a technology perspective but also from an operational /organizational perspective in which the system is deployed [118].

## Integrating existing knowledge sources and algorithms into the architecture

NLU at its core involves a number of mapping problems in order to achieve a level of understanding. From this perspective, medical language processing implies the development of mathematical models to represent language phenomena (*e.g.*, words, meaning, syntactic constructions) and the study of transformations that generatively map such constituents into higher order computer understandable representations that preserve meaning. A fundamental question then relates to exactly what mappings should be performed and how interpretable they should be. The design described in this paper is open to any implementations that satisfy the mapping tasks defined within the HSCM. A clear way to understand the role of existing NLP efforts is to view the NLU system from the Marr Tri-level perspective for complex information processing systems [49, 57, 119, 120]. This perspective includes the: 1) computational level (i.e., what problems the system is faced with, and levels of acceptable uncertainty); 2) algorithmic/representational level (i.e., how the problems can be solved, including for example, Bayesian methods, deep learning methods, and symbolic approaches); and 3) the physical level (i.e., how the system is physically realized). As preliminary work, we reviewed the general and medical NLP literature and conceptually organized NLP subproblems, algorithms, and knowledge sources along the Marr tri-level perspectives (see [58]). The review shows the relationship between the following items: the HSCM semantic layers, the state space of nodes within each layer, the mapping tasks between semantic layers, the sub-problems associated with each mapping class, the common knowledge sources employed within each layer, the typical algorithms and tools associated with various subtasks, and global optimization methods that can be employed. Thus, for example, Layers 0 and 1 of the HSCM identify word level semantics. The state space includes the inventory of all word level semantic descriptions. The subproblems associated with instantiating a node include: spelling correction, morpho-syntactic analysis, part-of-speech tagging, and assignment of word embeddings. The knowledge sources employed for these subtasks consist of probabilistic language models, medical idiomatic expression dictionaries, semantic lexicons, semantic selectional rules, and pre-trained deep learning transformer models. The algorithms and tools that could be employed for these tasks include clustering algorithms, regular expressions pattern matching, finite state machines, hidden Markov models, and neural network-based classifiers. The other layers of the HSCM can also be similarly viewed along these same perspectives. In summary, the HSCM is required to define the mappings for realizing a generative language-understanding framework. The execution strategies of these mappings can take on any best available approaches.

**Technical design evaluation metrics.** The evaluation metrics associated with the architecture can be linked to the described arguments concerning its conceptual advantages and disadvantages. A summary of possible metrics from various perspectives is summarized below.

**Human development effort point of view.** Since the HSCM should parallel how humans model the semantics of words, objects, events, and topics, an important metric is the cost associated with the effort by humans to construct the model. Various aspects per application might include the level of expertise, man-hours required for model development, and level of competence (e.g., errors and consistency) of HSCM authors in following guideline rules.

**From a knowledge engineering perspective.** Evaluation for building the HSCM knowledge base can be expressed in terms of traditional metrics used for ontologies and include expressiveness (representational adequacy), inferential adequacy (ability to infer new information), inferential efficiency, and acquisitional efficiency [121].

**From a software engineering perspective.** The HSCM defines a hierarchical graph based on semantic frames. Thus it embodies the ideas of an object-oriented organization for classes and their associated methods. Thus, the important object-oriented features of inheritance, abstraction, encapsulation, modularity, recursion, and procedural triggers can be easily realized by the architecture. Metrics associated with object-oriented software systems are defined in [122]. Performance metrics that can be defined from these features include time/effort to define new or edit existing HSCM nodes, and time/effort to debug definitional errors. Additional metrics related to the complexity of the HSCM graph include branching complexity, path complexity, data complexity and decisional complexity. These metrics become increasingly important as the breadth of semantic constituents and their complexity rises.

**From a computation perspective.** The general problem of NLU is difficult since it requires a mapping from all possible sentence inputs for a domain to all possible interpretations sanctioned by the software system. This results in a huge state space mapping. To tackle the "curse of dimensionality" issue, the system introduces structure in terms of hierarchical semantic composition. This allows the joint problem to be factored into a number of lower dimensional mappings. Computation time for the parser to search the HSCM for an optimal interpretation path is facilitated using a predictive coding algorithm. The search time savings is conceptually reduced from an exhaustive bottom-up search to a controlled hybrid search defined by only plausible hypotheses.

**Comparison of the HSCM and transformer model internal layers.** Transformer models such as BERT have become the state of the art for developing medical NLP applications [123, 124]. Deep learning models can generate these pre-trained encoder models in an unsupervised manner using vector based methods within a self-attention architecture [125]. Tenney et al. describes that when probing a BERT transformer model, one can discover that qualitatively the internal layers seem to be encoding raw language properties of input text such as part-of-speech tags, syntactic constituents, syntactic dependencies, semantic roles, co-references, and prototype roles [126]. The layers of the BERT model thus show some similarities to the HSCM layers. As in a traditional NLP pipeline, the lower levels of such encoder models emphasize local syntax, while the upper layers describe increasingly higher-level semantics. Autoencoders in deep learning methods have been shown to promote a hierarchical compositional representation to some degree [127, 128]. Although BERT does indeed show these abilities to identify various language-specific properties, relations and constituents, these mappings are made in a fuzzy statistical manner based on word associations using various self-attention mechanisms. In the case of the HSCM, the layering is based on a manually-specified semantic compositional view that reflects how human developers perceive language. The developers can precisely define the semantic granularity of the model that is useful for potential clients of the NLU application. For example, there is a general agreement on how one might create predicate-argument structures, or how a radiologist might define a semantic frame describing the properties of a mass (e.g., structured reporting forms such as BI-RADS [129]). In BERT models, there is no grounding of any constituents to real world ontologic definitions, although in some cases they can approximate this mapping [21]. Given the prolific applicability of BERT for many NLP problems however, it is clear that a sharable high quality language encoding knowledge source can be a core resource for many language-processing tasks. BERT was developed with the spirit of being a general language resource. The HSCM is being developed as a task

**Table 2. Comparison of properties of neuro transformer models versus the HSCM.**

| Aspect | Transformer Models | HSCM |
|---|---|---|
| Description of Layers | Can resemble a traditional NLP pipeline, with graded levels of semantic composition. Semantic constituents are coarse-grained. Highly dependent upon training corpus used and internal deep learning parameters. | Layers consist of a hierarchy of semantic types with ontological grounding. Constituents, in general, are fine-grained. Semantic composition of meaning consistent with human perspectives. Semantic abstractions can be high-level informational templates common to the medical informatics community (e.g., BiRads, RECIST) |
| Intended Use | General resource across diverse domains and tasks. | Tailored for each NLU task. |
| Semantic Granularity | Varies with training corpus; indeterminate. | Controlled by developers per NLU task |
| Effort | Data driven, unsupervised (BERT). (Not including decoding top-level classifier development effort per task). | Knowledge and data-driven, supervised. Substantial effort required in defining the semantic compositional hierarchy with associated grammars. Requires domain expertise. Development is progressive, benefiting from prior efforts. Parallel development can be relatively straightforward due to the localization of grammars to specific HSCM nodes. Standardization of methods for group development, however, will require community agreement. |
| Capabilities and Long Term Potential | Shows good performance for applications that require robust language sequence models. Concerns include a lack of ontologic grounding and awareness of real-world knowledge (e.g., discourse models, situational micro-theories, and clinical context). It is unclear what the necessary parameterization of a network should be to ensure it works for a growing number of medical NLU tasks. | Framework conceptually has the potential for interpreting the intensions of authors by incorporating expectation models for targeted clinical communication topics. Concerns include the level of development effort and integration of knowledge sources into the representation. |
| Adaptability / Configurability | Transformer encoder models such as BERT are static in the sense that the structure and parameters do not change once they are trained. They are computationally expensive to train limiting the pool of individuals/organizations that can generate such a model. | The HSCM is used in a dynamic fashion depending upon the global contextual specifications of the NLU task, and the upward activation patterns (i.e., dynamic routing) fired during the predictive coding steps (i.e., performs adaptive computation depending on local and global contexts). |
| Transparency | Relatively opaque; not uncommon for tasks to utilize spurious correlations in data for features. | Explanations realized from paths through the HSCM for a given parse of a text excerpt. |

specific resource. Table 2 compares some of the properties associated with the HSCM model versus popular deep learning transformer models such as BERT.

**Efficiency mechanisms.** Computational efficiencies of the HSCM design are mainly achieved from manifestations of semantic composition (representational efficiency) and hierarchal predictive coding (processing efficiency). Imposing a compositional structure (i.e., factorization) is known to contribute significantly to reducing the dimensionality (i.e., computational complexity) of the parsing problem [130, 131]. Efficiencies are gained by factoring the overall NLU problem into a number of lower-dimensional mappings. A compositional structure provides a framework for 'part sharing' which allows development to proceed in a piece-wise systematic way. This part sharing strategy can lead to an enormous reduction in computational complexity [132, 133]. Predictive coding offers processing efficiency since only plausible hypotheses specified within the HSCM need be tested. A combination of bottom-up (hypothesis formulation) and top-down (hypothesis testing) processing conducted within a hierarchical predictive paradigm greatly reduces the search state space for a viable global sentence parse. Note that a purely bottom-up (inverse problem) approach to semantic parsing is regarded as an ill-posed problem [134]. The HSCM model provides semantic compositional constraints to reduce the number of possible interpretations. A full theoretical discussion of how predictive coding can readily solve high-dimensional mapping problems (e.g., all possible input signals to all possible interpretations) using the free-energy theory" can be found in [135]. Worth mentioning is the relation of predictive coding to backpropagation learning and its efficiencies as employed by neural networks [136, 137].

**Importance of compositionality.** Compositionality for language understanding is central in our design on the grounds of two long-standing principles in linguistics: 1) Bottom-up: Principle of Composition–that the meaning of the whole sentence is a function of the meaning of its parts [31, 68, 69, 138]; and 2) Top-Down: Context Principle–that words have meaning only as constituents of the sentence [138]. Fillmore described language understanding from the perspective of semantic frames and the idea that contextual regularities can be encapsulated in a grammar [139]. Our design incorporates these ideas by proposing the use of semantic frames for all levels of tokens, attaching grammars for each constituent within the HSCM, and hierarchical incremental parsing to improve context within each processing stage. Fig 4 shows the incremental synthesis of semantic constituents that are aggregated into a unifying sentence-level semantic frame. For a general discussion of computational efficiencies gained from compositional factorization, see [130, 132]. Additionally, there has been much discussion within the AI community related to what types of knowledge are required for systems to truly generalize beyond their training data. Central to these discussions is the need for compositionality [35]. Further discussion regarding the benefits of compositionality for NLU can be found in [131], including its benefits with respect to annotation consistency.

**Balance between fine-grained comprehension and general applicability.** Inference models generally experience a familiar trade-off between accuracy and robustness (e.g., recall vs. precision) [140]. A number of design compromises need to be considered for each given task. These considerations include a) performance requirements of the driving NLU task (e.g., error rates, semantic granularity, b) the need for an explanation of answers, c) the types of errors observed (e.g., similar to humans), d) processing speed, and e) text coverage. The weighting placed upon such considerations often depends upon whether the NLU task is population-centric or patient centric. The population class includes medical applications that aim to estimate or improve upon a population parameter. Examples of tasks that prioritize breadth of coverage (i.e., generality) include the identification of patients who match inclusion criteria for assembling teaching cases and discerning patients who are possible candidates for a specific clinical trial. General-purpose language knowledge sources such as BERT can be quite effective in improving targeted performance parameters (e.g., percent of patients enrolled in a clinical trial). If the pool size of the patient population is large, the expectations of an NLU application may allow tail distribution samples to suffer from relatively poor performance. That is, it may be acceptable to have a relaxed expectation of accuracy for rare/difficult language use. However, there are also cases, such as in identification of patients with rare conditions, where it is important to identify specific criteria and/or infer target cases based on causality. In such situations, a system that outputs rich semantics and/or infers causal meaning might be more effective. The second class of problems to consider is patient-centric NLU applications. While robustness across all expected note types, authors, and institutions is desirable, sacrifices in accuracy can be highly detrimental to long-term clinical acceptance. This can be especially true if blatant errors are experienced in, for example, point-of-care applications or patient treatment planning meetings such as tumor boards [141]. To avoid such issues, precise micro-theories that can supplement the required context for text comprehension may be necessary. Such fine-grained modeling can impose real-world semantic constraints on meaning representations [95]. Until various levels of clinical evaluation are performed [115, 117], it is often difficult to estimate the required performance parameters of a task until several rounds of efficacy and clinical outcome studies have been performed. This is most evident in the fact that there are a large number of technical evaluations reported in the medical NLP literature, yet the number of implementations in actual clinical use with reported clinical value remains scarce [116, 142, 143]. Transparency of algorithms and patient safety concerns remain as critical concerns in this regard [142, 144].

**Summary of main arguments in favor of a cognitive framework for medical NLU.** This is a difficult question because in making a decision about strategic directions, one must carefully evaluate the growth potential of alternative systems and project which paradigm can best serve as a long-term framework for efficiently deploying medical NLU applications at the highest possible standards, including maximizing patient benefits, minimizing patient harm, and minimizing cost to society.

The question of whether a data-driven, knowledge-driven, or hybrid system should be the driving paradigm for language understanding has been lively debated for many years [35, 145, 146]. The two paradigms vary significantly in many respects. Deep learning system development is data-driven, relying of manipulating numeric representations that are continuous. Cognitive system development is largely knowledge-driven, relying on the manipulation of symbolic representations that are discrete in nature [147].

Much of the discussion regarding the strategic direction to follow centers on the degree to which prior knowledge is required for language understanding, as for example, argued by the rationalist versus empiricist views [148]. In principle, there is agreement that NLU systems need declarative knowledge in order to achieve human levels of understanding [149]. The differences in the two paradigms relates largely to how this knowledge is to be acquired and represented within a software system. A few issues to consider are presented below.

**Amount and acquisition approach of knowledge.** The sheer amount of estimated knowledge can deter what strategic directions one follows. In deep-learning data driven methods, it is typically assumed that the goal of a knowledge base is to serve as a foundational language resource for a broad spectrum of applications [150]. These foundational models thus assume that the application space and associated text are open-ended and ambiguous and that it is not feasible to specify the required knowledge using manual or supervised methods. Self-supervised methods such as autoregressive self-learning (e.g., GPT-3) and auto-encoding self-learning approaches (e.g., BERT) are commonly used. The assumption is that the knowledge will emerge automatically by analyzing a large amount of text using such self-learning algorithms. The base "genetics" supplied to these algorithms that dictate how the knowledgebase will evolve from no structure to highly structured (billions of parameters) is, surprisingly, a simple set of rules that are applied iteratively to the training corpora [151]. The mottos of "attention is all you need" [125] and "scale is all you need" [152] encapsulate the ideas of how such foundational models are realized.

Conversely, the cognitive approach seeks to adapt or incrementally acquire knowledge on an application-by-application basis. The assumption is that, given the limited scope of each task, it is feasible to manually specify over time a comprehensive metaphysical logical representation of the essential information content required by a driving application. It further assumes that this representation will be relatively stable and can evolve incrementally. The approach emphasizes meaning by grounding semantic constituents to real-world interpretations. New applications are supported by either utilizing views of the model already developed (i.e., part sharing) or adding/modifying new components and linkages to the overall representation.

*Quality of knowledge.* The data-driven approach emphasizes breadth of knowledge, attempting to extract whatever regularities can be inferred using the attention-based rules ingrained by self-learning algorithms. While the breadth of knowledge captured by pre-trained deep learning models appears substantial, their quality is indeterminate. Because of their emergent behavior, deep learning models are hard to understand and control. The quality of the knowledge depends on the locality rules for attention (i.e., close to broad), the training text (e.g., amount, type, and order), the model structure (e.g., network size), and training protocol. While the quality of knowledge captured in pre-trained models such as BERT appears surprisingly exceptional based on their remarkable successes, it is not uniform across the spectrum of

knowledge elements intrinsic within the text they are trained on. Current models are designed to solve the language masking problem, which may be unrelated to an applied downstream task. These models seem to be able to learn some types of regularities in language rather accurately (e.g., syntactic relationship) but poorly at others (e.g., temporal reasoning tasks) [153]. Karlgen and Kanerva discuss the theoretical issues limiting the semantic accuracy and semantic similarity abilities of high-dimensional vector representations [154]. Global and irrelevant statistical dependencies can blur local intrinsic relevant features in high-dimensional representations, as noted when computing multi-dimensional centroids (a form of lossy compression). Composition within a deep learning architecture can further exasperate the semantic quality of such latent representations, entangling concepts in a spurious manner [155, 156].

The cognitive approach emphasizes high-quality knowledge that is consistent with views of how the problem domain should perceive the world situation. The semantic granularity of a cognitive model is under the developer's control. The modeling process ensures at a logical level that the necessary content for explanation is included. It focuses on including only enough information required to understand the text to support the NLU actionable response. The definition of the logical model however is subjective and is based on the views of the developers and/or adapting communities. In general, it may require numerous iterations to be comprehensive for the target application over many site deployments. The quality of the knowledge to be included is very specific to the micro-world associated with an NLU task in order to address difficulties such as coreference resolution, clarification of ellipsis, interpretation of coordinating conjunctions, and proper assessment of event temporal order. This ontologic grounding of meaning provides the key knowledge substrate for debugging, transparency and explanation. A valid question to be raised about the cognitive paradigm is whether such a comprehensive model can be pre-defined for a given application, and what are the dangers of incomplete or erroneous constraints within the model. Such deficiencies can limit the performance of an NLU application and have a significant impact on the performance of unseen samples [145].

*Integration of external knowledge*. Current transformer models do not have specific mappings to ontologic concepts. Their distributed multi-dimensional representation makes mappings to a specific user-defined view of the domain difficult. The cognitive view emphasizes ontologic representations at various levels of semantic abstraction. Integrating external knowledge sources is conceptually possible to order to increase the scope of queries supported by the knowledgebase. Common sources include thesauruses (e.g., WordNet and UMLS), logical definitions of predicate-argument structures (e.g., PropBank [60, 62]), and numerous medically topic-specific ontologies (see, for example, the compilations at The Open Biological and Biomedical Ontology Foundry [157]. At a practical level, integration of heterogeneous ontologies can be challenging due to the standardization of interfaces at both the logical model and processor levels. This overhead is commonly seen in issues related to mismatches in syntax, intended use, node definitions, label ambiguity, and inheritance complexities [158].

*Is required knowledge known to humans*?. A more basic question related to knowledge inclusion is whether or not it can be specified. That is, if it can be formally specified, it can be theoretically implemented in software. For example, deep learning systems for image and signal analysis domains have been able to reach high levels of recognition accuracy because they have the potential to detect complex imaging patterns (e.g., textures, hierarchical layered, periodicities, and self-similarity features) that may not be obvious to human observers. In language, however, humans have an inherent ability to identify relevant content for almost all language understanding tasks, and there is a long and rich academic history of defining representations for language [96]. The point here is that, although we have the practical knowledge to specify a comprehensive semantic substrate for inferring meaning intent for a given NLU

task, the trend is to avoid such unfashionable building of this logical symbolic layer through manual means.

*Computational science fields vs. medical informatics culture.* The direction of medical natural language processing research has been significantly influenced by the academic culture of traditional computing fields such as computer science and statistics. The computational disciplines value algorithms that are generally applicable to a wide scope of problems, with a balance of effectiveness and efficiency. The estimates of time and space complexities of an algorithm are valued with the assumption that the algorithm will perform complex operations on large amounts of diverse input data. Manual specification of domain knowledge has been traditionally discouraged by criticisms related to algorithm generalizability. Annotation of training examples, which may require domain expertise, is commonly viewed as "*tedious*," "*expensive*," and/or "*extremely time consuming.*" This places a high value on unsupervised methods. Feigenbaum comments that this reluctance to avoid domain knowledge is likely related to the skill and interest boundaries between computational-oriented experts (e.g., computer scientists and statisticians) and domain-application experts (e.g., medical informaticians and clinicians) [159]. Within our applied field, adapting these biases simply perpetuates the theory-practice chasm, potentially limiting the abilities of NLU systems to achieve a human-level of comprehension. There is a tendency because of this bias to take a "do nothing" approach. Computationalists, we might say, tend toward being generalists, with the goal of applying algorithms to a broad class of data. Informaticists focus on tasks, thereby, operating within the mindset of a specialist, investigating all aspects of a problem in all use scenarios to strive toward a perfected product. The criteria for a good algorithm (e.g., generalizability) are not necessarily applicable to the criteria for a good application. Without a specialist demeanor, an NLU application likely will not survive within a clinical environment as it must perform and be managed according to the needs presented within the realities of a clinical ecosystem. The medical informatics community ultimately values a system that facilitates medical care, regardless of whether a particular solution is computationally fashionable. We cannot selectively filter which issues brought up by users of an NLU application to ignore based on the limitations of preferred methods. We cannot ignore complex language understanding phenomena that may exist in the data because there is no theoretical framework for intentional inference or because of an unwillingness to put effort into solutions that require manual effort. As an applied field, we believe the development of rich medical domain-specific models, which provide the basis and transparency for interpretation, should be promoted. The medical informatics community, in fact, has a long history of enthusiastically pursuing the construction of fine-grain data models and ontologies. Friedman had previously discussed the merits of building sublanguage models for improving the semantic granularity of medical NLP system outputs [160]. Given the relative stability of the concepts, predicates, and communicative goals of a given task, we speculate that these models should be realizable with diligent and persistent efforts from knowledge engineers.

**Hybrid neuro-symbolic directions.**    Deep learning is a highly active, rapidly changing field. New directions that emphasize learning compositional models of real-world objects and events are being investigated in order to acquire more human levels of cognition [149]. Hybrid systems that borrow from the strengths of symbolic and deep learning paradigms are being actively pursued [155, 161, 162]. Leading AI experts have acknowledged the need for NLU systems to integrate knowledge at all levels of comprehension [149, 163]. Google Search, for example, uses both a deep learning BERT model and a symbolic knowledge graph to disambiguate word sense. Commonsense knowledge inferred using deep learning methods is an active area of research [164]. Symbolic systems have the advantage of symbol grounding from which various types of logical inference can be performed. Symbolic systems are at risk for lack of

coverage and/or context-specific errors in their structural and semantic specifications. Deep learning systems have the advantages of learning complex semantic abstractions as well as contextualizing word/phrase used over broad coverage. Generalizing grammar patterns and/or improving semantic activation within a cognitive paradigm can be significantly supplemented using deep learning features [165]. Dynamic agents can easily then combine symbolic features (e.g., syntactic-semantic word patterns) and deep learning features (e.g., word and graph embeddings) to generalize compositional grammars and/or semantic activation triggers for hypothesis testing and generation. A comprehensive discussion of general arguments in support of a hybrid paradigm for AI is given in [166].

## Conclusion

The strategic direction to pursue for medical NLU is a topic that has not been thoroughly discussed. Many have already conceded to the direction of deep learning architectures. However, many of the arguments and biases of a data-driven deep learning approach stated in the general computing field do not necessarily hold within the medical informatics application field. Medical NLU problems typically do not require processing huge amounts of data within a limited time. Medical informatics endeavors do not find it difficult to seek the collaboration of domain experts, but rather always work closely with them. The medical informatics community is not often deterred from constructing comprehensive knowledge sources. It has a long rich history of building metaphysical representation of various medical/biological phenomena. Data-driven claims that applications are "ephemeral" are not applicable [see The Data-Centric Manifesto–datacentricmanifesto.org]. Medical NLU tasks are motivated by real needs that have relatively stable specifications. Clinical failures are primarily due to implementation issues and not specification of needs [58, 117]. Our position is that the knowledge driven cognitive paradigm better address a number of theoretical and practical concerns of data driven methods. The cognitive approach allows each NLU task to define its required level of semantic content and granularity. It defines an organic transparent semantic substrate to support logical inferencing and from which explanations can be derived. It provides a means of integrating various existing ontologies to extend its coverage and inferencing capabilities. Semantic composition allows constituent grammars to be defined locally to each HSCM node, thereby facilitating community development. Composition can also simplify training efforts using grammar-based semantic annotation schemes, which become increasingly important with the complexity of the NLU task [130].

In conclusion, we present arguments for an NLU architecture that is cognitively inspired. At its core is the HSCM that imposes structural constraints on the expectations of how information is expressed in the targeted language domain. This applied structure allows the system to process input sentences using a predictive coding paradigm. An agent based processing scheme allows various algorithms and inferencing modes to be available for a given NLU subtask. Although we acknowledge that a number of alternative architectures are possible, we believe that this framework has the potential for accommodating important design considerations including:

1. Theory–a foundational architecture should be able to accommodate the best theories of language understanding from linguistics, cognitive science, and neuroscience;

2. Computation–the framework must accommodate the most recent advances related to computing the most likely interpretation (in an information-theoretic sense) for a given text input;

3. Flexibility–the design needs to be adaptable, allowing for different algorithmic approaches to be explored;

4. Transparency / Explainability–It is desirable for an NLU system to be transparent as to how it is making its decisions. The model should be able to explain how it derived its final interpretation in terms of only sanctioned (sub) interpretations as defined by the HSCM.

5. Applicability–the architecture must be applicable to diverse domains/applications that may require different degrees of accuracy and coverage and processed in a timely manner;

6. Interoperability–the logical compositional model should ultimately be able to semantically interoperate with other knowledge sources (e.g., causal models of disease and various ontologies) in order to perform higher-level inferences at any level of interpretation.

7. Scope / Scalability–The architecture should have a high growth potential, evolving into a high-density system of nodes and connections that can be utilized to understand a greater scope of sentences within a greater range of application contexts. The architecture should be able to build upon existing efforts in a theoretically principled and unifying manner.

Implementation of a prototype system for analyzing descriptions of tumors from radiology reports has been ongoing within our department and has been the driving application for developing many of the ideas of this design. Details of this implementation are planned to be reported in the near future.

## Acknowledgments

The author would like to thank the many members of the UCLA Medical Imaging Informatics Graduate Program for their constructive discussions related to the topic of this paper. We would also like to thank Lew Andrada for grammar and style related edits.

## Author Contributions

**Conceptualization:** Ricky K. Taira, Anders O. Garlid, William Speier.

**Investigation:** Ricky K. Taira.

**Writing – original draft:** Ricky K. Taira, Anders O. Garlid, William Speier.

**Writing – review & editing:** Ricky K. Taira, Anders O. Garlid, William Speier.

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
