## [Decision Letter · Decision Letter 0]

6 May 2021

PONE-D-20-39301

Design considerations for a hierarchical semantic compositional framework for medical natural language understanding

PLOS ONE

Dear Dr. Taira,

Thank you for submitting your manuscript to PLOS ONE. After careful consideration, we feel that it has merit but does not fully meet PLOS ONE’s publication criteria as it currently stands. Therefore, we invite you to submit a revised version of the manuscript that addresses the points raised during the review process.

We look forward to receiving your revised manuscript.

Kind regards,

Ilya Safro, Ph.D.

Academic Editor

PLOS ONE

Journal Requirements:

'This work was funded in part by funds from the National Institutes of Health grants R01-CA226079, R01-LM012309, R01-CA1575533, R01-LM011333, and U24-AI117966.

RKT was the recipient of funds from these sources in the role of an investigator.  The funders had no role in study design, data collection and analysis, decision to publish, or preparation of the manuscript.'

a. Please provide an amended statement that declares *all* the funding or sources of support (whether external or internal to your organization) received during this study, as detailed online in our guide for authors at http://journals.plos.org/plosone/s/submit-now

Please also include the statement “There was no additional external funding received for this study.” in your updated Funding Statement.

Reviewers' comments:

Reviewer's Responses to Questions

**Comments to the Author**

1. Is the manuscript technically sound, and do the data support the conclusions?

Reviewer #1: Partly

Reviewer #2: Yes

Reviewer #3: Partly

2. Has the statistical analysis been performed appropriately and rigorously? 

Reviewer #1: N/A

Reviewer #2: N/A

Reviewer #3: N/A

3. Have the authors made all data underlying the findings in their manuscript fully available?

Reviewer #1: Yes

Reviewer #2: Yes

Reviewer #3: No

4. Is the manuscript presented in an intelligible fashion and written in standard English?

Reviewer #1: Yes

Reviewer #2: Yes

Reviewer #3: Yes

5. Review Comments to the Author

Reviewer #1: The paper described design of a semantic model and an associated parser used to transform a free-text sentence into a logical representation of the meaning of clinical text. The main contribution of this paper is to enhance machine understanding clinical text semantically. While the four-point framework aims to provide a structure for designing NLU system, some points had relatively similar concepts in natural language processing, e.g. both “Predefined Semantic Representations” and “Semantic Activation Networks” are both similar to knowledge models in medical knowledgebases and ontologies. Overall, the paper lacks comparison to other NLP concepts, tasks and systems.

Besides, there are several major problems:

1. The structure of the paper is confusing, “we first introduce the overall NLU problem highlighting the need for a predefined compositional structure” – this should be defined in the Introduction.

2. The author should also highlight major deliverable of the paper “Hierarchical Semantic Compositional Model (HSCM)” earlier in the manuscript as well as in the abstract.

3. Definition of “surface words” is not clear.

4. How is Semantic Layer 4-6 different from each other?

5. “This structure enables a more efficient process of encoding sentence meaning, by facilitating a generative model.” – how do you measure efficiency? Do you compare your parser with other semantic parsers?

6. “Our design involves explicitly defining this structure in a way that parallels the manner in which human compose meaning.” – how is your methods different from other NLP tasks that extract information from unstructured clinical text and create a structured output.

Reviewer #2: This manuscript proposes a new framework for NLU (natural; language understanding) based mostly on semantic analysis (semantic memory, composition, activation and hierarchical predictive coding). This interesting framework is based on a few human cognition characteristics and contrasted with current deep learning-based approaches. The manuscript is very well written, but the framework description remains quite vague in several aspects and leaves most details and possible implementations to be defined. This vagueness might keep it more flexible and technology-agnostic but also makes understanding several aspects of the proposed framework quite difficult.

The following revisions are recommended:

Major Compulsory Revisions:

1. The framework proposes an approach that is presented as applicable to all clinical narratives and their possible content, but also mention that circumscribing the scope of this possible semantic content is part of the modeling process. This is a critical issue and balance to find between a very specific semantic field specific to one specialty, note type, institution and even individual author, and broad biomedical and general knowledge. More discussions around this issue and strategies to address it should be added.

2. As mentioned above, the complexity of certain aspects of the proposed framework and the vague description make understanding sometimes difficult. A few punctual examples are provided, but a more complete and realistic example (maybe broader than the radiology report examples provided) used throughout the different steps of the proposed NLU process would significantly help understand the concepts proposed.

3. Among desirable characteristics of NLU/NLP frameworks, generalizability is among the top and is a very current problem, but mostly ignored in your manuscript; it is not even mentioned in its list of qualities. On page 11, you state that you favor topic-centric and corpus-based approaches (with specific note types); this strategy might allow for better performance in this specific field, but also tends to generalize to other domains with difficulty.

4. Word sense disambiguation is an important step in the semantic analysis of clinical text, and how it could be implemented remains unclear. Would it be part of semantic layer 1 (e.g., for abbreviations expansion)? Or another one for eponyms for example? Similarly, analysis of the local context of the information/concepts identified (e.g., negation) is critical but not clearly part of your semantic constituents; layer 4? Or 5?

5. The discussion contracting the proposed framework with deep learning-based approaches using word embeddings etc. is very important nowadays and should be expanded with more structure and concrete examples. Demonstrating the proposed superiority of your approach would rely on it.

Minor Essential Revisions:

1. Stating that automated deep understanding of clinical notes remains elusive and generally far from human abilities is probably correct, but requires evidence (e.g., cited bibliographic references) and would benefit from a more detailed description of specific limitations your proposed framework would address.

2. On page 5 last line, “lexical level” would match the other levels mentioned better than “at the level of words”.

3. On page 32, stating that the clinical language is more restrictive than “open-ended” text might be true for radiology reports but has been demonstrated as less restrictive in other domains and note types. Some clarification is needed.

Reviewer #3: It is refreshing to see described, a proposed system design drawing from well-researched theories in the cognitive sciences, and practical considerations in the medical imaging reporting domain, that attempts to performs knowledge synthesis and inference similar to specialized human experts. This is in sharp contrast to many leading conference and medical NLP workshop papers in recent times employing the latest state of the art deep learning methods, but all of which read very much the same.

This work could definitely be poised to become a unique and important contribution in the area of medical NLU, however there are some key aspects of completeness that must be addressed first. The main thrust of the article seems to be around the premise that "compositionality" plays a key role in the human-like understanding by machines, of complex sentence structure, stating perhaps rightly that we would not lose any information given in the sentence by factoring it into components that are themselves meaningful at various levels of semantic abstraction. These components seemingly could even be at the level of ontological primitives and propositions besides characters, tokens or phrases. However, little is presented in terms of evidence or experiments to support this claim or hypothesis.

The introduction and background sections give the reader a solid overview of the different elements that will go into the design of the framework, such as semantic composition, semantic activation networks and hierarchical predictive coding. However, the paper structure, while it is meant to be about the design considerations for such a framework, lacks the structure of a clear hypothesis and supporting (even preliminary) experiments, analysis, or initial prototype, that serves to "inform" such a design.

It is good to see models such as BERT, BioBERT cited in references, however the necessary comparisons as to how the proposed framework is at least "different" if not better, than these, by way of just some simple examples or analysis is lacking. E.g. if the authors are familiar with the recent work "BERT rediscovers the classical NLP pipeline." by Tenney et al. that source could have been cited and appropriate comparisons presented. E.g. this aforementioned paper talks about how internal BERT architecture layers are found to perform increasingly complex tasks, viz. part-of-speech tagging, parsing, entity recognition, semantic roles, and finally coreference, in that order, towards NLU. This is reminiscent of the various level managers in Figure 4 of this draft. Thus a simple comparison by means of a chart or a table between the framework in Figure 4, and BERT, could have been provided, highlighting side-by-side, how existing language models of the day such as BERT, or its fine-tuned variants like BioBERT process and understand language at each level, compared to the proposed HSCM framework. This might give the reader an immediate intuition of how and why HSCM can be expected to perform better and perhaps produce more human-like inferences. At the least it would highlight the benefits and drawbacks of each paradigm. Also, while the implementation details of each piece of the framework has been left to the user, it is not clear what sort of time complexity considerations are at play, for encoding and decoding/inference functions of the framework, e.g. steps 1,2,3 and 4 of Figure 5. This may lead the reader to assume deep learning-based systems are at a greater advantage, at a fraction of the representation, training and inference cost. Some estimate of what time and space complexity each layer/piece of the framework involves by way of some simple example, might give the reader a better idea.

The claim of greater interpretability possible with the HSCM framework and how it arrives at final decisions in terms of "true intended meaning" of a parse has not been entirely elaborated either. Also, the authors seem to suggest that HSCM can better handle out-of-vocabulary concept formulation and meaning understanding by way of semantic activation network. An illustration of this can again be provided via a comparison table or graphic visualization using a simple/concrete example that shows how HSCM is expected to handle an unseen term, "concept", or "semantic abstraction" versus skip-gram word2vec, for example. This is an opportunity to highlight how HSCM can provide superior human-like understanding compared to simple inference using word2vec. Perhaps HSCM is able to leverage embedding-based representations at its various levels, which gives users of the framework best of both worlds. This aspect could also be elaborated on. Finally, I believe that the work should be presented for publication after some very basic observations and outcomes are gathered and evaluated from an initial prototype. Overall, the paper outlines some powerful ideas that seem to have the potential to "truly" solve NLU for the medical domain, however it must be presented in the context of what current medical NLP/IR systems based on deep-learned representations do not yet have the capacity to solve, and by showing via example or anecdote from the HSCM framework, as to how a system based on this framework has a better chance of solving that specific NLU problem.

6. PLOS authors have the option to publish the peer review history of their article (what does this mean?). If published, this will include your full peer review and any attached files.

Reviewer #1: No

Reviewer #2: No

Reviewer #3: No

---

## [Author Response · Author response to Decision Letter 0]

17 Nov 2021

We have included responses to each concern of all the reviewers in the "Response to Reviewer" document.

---

## [Decision Letter · Decision Letter 1]

19 Jul 2022

PONE-D-20-39301R1Design considerations for a hierarchical semantic compositional framework for medical natural language understandingPLOS ONE

Dear Dr. Taira,

Thank you for submitting your manuscript to PLOS ONE. After careful consideration, we feel that it has merit but does not fully meet PLOS ONE’s publication criteria as it currently stands. Therefore, we invite you to submit a revised version of the manuscript that addresses the points raised during the review process.

We look forward to receiving your revised manuscript.

Kind regards,

Ilya Safro, Ph.D.

Academic Editor

PLOS ONE

Reviewers' comments:

Reviewer's Responses to Questions

**Comments to the Author**

1. If the authors have adequately addressed your comments raised in a previous round of review and you feel that this manuscript is now acceptable for publication, you may indicate that here to bypass the “Comments to the Author” section, enter your conflict of interest statement in the “Confidential to Editor” section, and submit your "Accept" recommendation.

Reviewer #1: (No Response)

Reviewer #4: (No Response)

2. Is the manuscript technically sound, and do the data support the conclusions?

Reviewer #1: Partly

Reviewer #4: No

3. Has the statistical analysis been performed appropriately and rigorously? 

Reviewer #1: N/A

Reviewer #4: N/A

4. Have the authors made all data underlying the findings in their manuscript fully available?

Reviewer #1: Yes

Reviewer #4: Yes

5. Is the manuscript presented in an intelligible fashion and written in standard English?

Reviewer #1: Yes

Reviewer #4: Yes

6. Review Comments to the Author

Reviewer #1: 1. In "Dicussion", the author mentioned that clinical text "varied with respect to their formality, style, and flow". However, it is not clear how this varaibility bring problems to existing NLU systems.

2. In "Speaker-Listener Model" section, it is not clear what the model is. Is it a mediator between Speaker-Listener or something else. This paragraph is confusing.

3. The example provided in the "Wrong use of valid terminology", I don't think this is a good example, it is just a non-standardized unit measure and a standardized unit quantity can be calculated.

4. "Punctuation Mis-use" was mentioned as examples of language complexities in medical reports. However, does this bring problems to NLU system at all? Normally NLP systems would remove punctuations at a certain stage of data processing.

5. Example in "Temporal Ambiguity" is also not really ambiguity. It has to do with information not captured by the physiciian.

6. "Inferred information from practice guidelines" is not a examples of language complexities, but rather a knowledge complexity which will require periodic update of the knowledgebase or proper version control. I think it should be discussed separately as an essential part of system design.

7. The author compared Cognitive Framework with deep learning based framework for NLU. While I do see the similarities between the two frameworks in terms of hierachical structure, a more fair comparison would be with other previous content-based systems (or there can be other names used e.g., scenario-specific systems, systems enabling spatial and temporal constraints).

Reviewer #4: This paper describes a concept design of a hierarchical semantic compositional framework to provide an internal model for guiding the interpretation process. Overall, this concept paper reviews some ideas that might be useful for achieving a “deep and true” understanding of clinic texts, however, it lacks a detailed review for the applicable contexts and tasks. For a successful publication, I believe that this paper shall provide either theoretical or empirical contributions. I would strongly suggest that it shall include empirical support to justify the performance of the proposed framework. Also, developing metrics for the proposed framework’s evaluation is important and human experts’ evaluations seem to be also essential in this matter. However, this paper only presents a general summary of evaluation required by medical NLU systems in P35. It is uncertain for the HSCM’s evaluation and so to its real effectiveness.

Main concerns:

Introduction

Research motivations:

The authors stated “… NLP/NLU of clinical reports is an important area in medical informatics … However, it is challenging to perform deep understanding of clinical notes by computers …” without giving a clear definition of “deep understanding”, what does it mean? Since this is the motivation of designing such framework, it is necessary to define the level of understanding the filed expects rather than simply saying “closer to human capabilities”.

Also, “true intended meaning” in such context lacks a clear definition. It will be good to give examples for some texts derived from the clinic reports and show both “wrong interpretations” VS “true intended meaning”.

Design:

The HSCM is designed for a sentence-level understanding, what is the reason for picking this level? Semantic understanding is important as stated in the Background section, so defining the semantics used in the HSCM model is essential for its success. Performing pilot studies to test the length of words will be helpful in this matter. The sentence-level model processing might not be an optimal selection, because the clinic texts/reports might be very context-sensitive, such as a good understanding only can be achieved via reading several paragraphs, thus, there might be some misunderstanding due to the selection of the unit.

The introduction section lacks a statement of findings and a summary of contributions.

Medical NLP studies have lots of applications. This paper in general does not include an overview of the NLP/NLU systems, which is especially important for conducting a systematic knowledge of the NLP concepts, tasks, performance in the medical domain.

Definitions for performance metrics are missing. This paper proposes a framework (a concept paper) without conducting any experiments using “real clinic data”, so the argument “This structure enables a more efficient process of encoding sentence meaning, by facilitating a generative model.” is not supported from the overall study.

It is not clear that the innovative aspects of the HSCM, how does it different from other NLU methods that could effectively process clinic texts? As the authors stated, unstructured clinical texts need to be carefully handled to create a structured output, but why HSCM could potentially outperform other existing systems? Here is a lack of justification.

Also, any potential implications and applications will be further developed based on the HSCM framework?

Figures:

Figure 4, describing the model architecture with various level managers, has many abbreviations, however, there is a lack of explanation for their meanings.

Minor concerns:

Writing:

There is an underline for the word “understanding” on P3 line 55, which looks weird.

Please insert figures to the corresponding places of the manuscript. The authors put all the figures at the end of the manuscript which causes large inconvenience for reading.

7. PLOS authors have the option to publish the peer review history of their article (what does this mean?). If published, this will include your full peer review and any attached files.

Reviewer #1: No

Reviewer #4: No

---

## [Author Response · Author response to Decision Letter 1]

17 Sep 2022

Design considerations for a hierarchical semantic compositional framework for medical natural language understanding

Response to Reviewers

The authors would like to sincerely thank the reviewers for their efforts in providing critical and helpful comments for this paper re-submission. Below we address each point of concern from each reviewer.

Reviewer #1:

Q-01: In "Discussion", the author mentioned that clinical text "varied with respect to their formality, style, and flow". However, it is not clear how this variability bring problems to existing NLU systems.

This statement on page 32 was made in relation to understanding the knowledge expectations and inference strategies that would be required for deciphering the meaning of clinical text when presented with different language styles and levels of semantic complexity. NLP models based on word sequences (e.g., topic models, Markov chains, and deep learning transformer models) are applicable to all grammatical styles and topic areas, but have limitations with respect to their ability to ground meaning to real world concepts. The degree to which NLU systems can infer intended meaning vary with respect to text writing styles. 

For example, radiology reports tend to be authored using well-formed grammar in a detailed declarative writing style and, as such, are relatively straightforward to process [Langlotz 2014]. Conversely, physician notes often present a telegraphic style of writing in which a number of abbreviations and word sequence fragments are observed [Stallinga 2015]. In such cases, the main points of the communication can be difficult to infer due to lack of details that must be inferred based on the clinical context. With respect to the narrative flow, reports such as discharge summaries can include complex descriptions of the patient’s episodic timeline, that require information over many sentences to be comprehended and connected. In such narrative reports, various difficult language aspects are common, including coordination, coreference resolution, and ellipsis.

Langlotz CP. The radiology report: a guide to thoughtful communication for radiologists and other medical professionals. 2015, ISBN: 978-1515174080.

Stallinga HA, ten Napel H, Jansen GJ, Geertzen JH, de Vries Robbé PF, and Roodbol PF. Does language ambiguity in clinical practice justify the introduction of standard terminology? An integrative review. Journal of Clinical Nursing. 2015;24(3-4):344-52.

We have added text in the Discussion section to clarify these points.

Q-02: In "Speaker-Listener Model" section, it is not clear what the model is. Is it a mediator between Speaker-Listener or something else. This paragraph is confusing.

The “Speaker-Listener Model” is a long-standing model for language comprehension that describes how information might be conveyed from a report writer to a report reader in the presence of a noisy channel that might include inaccurate and/or incomplete signals (e.g., words). It describes how a listener and reader require some form of synchrony in terms of the expected communications and context from which the dialogue is made [Stephens 2010]. With respect to medical reports, we see a wide variety of language styles (formal English, terse sentence fragments, etc.) that are unambiguous to the reader due to the common understanding between author and reader regarding the clinical context (e.g., procedure type, prior communications) and general domain knowledge in which the observations, events, and recommendations were made. The point we make is that such an expectation model may potentially be required for comprehending such abbreviated patient reports. We have included the reference below to provide the reader with a link to pursue further details on this concept.

Stephens GJ, Silbert LJ, and Hasson U. “Speaker-listener neural coupling underlies successful communication,” PNAS 107(32):14425-14430, 2010.

Q-03: The example provided in the "Wrong use of valid terminology", I don't think this is a good example, it is just a non-standardized unit measure and a standardized unit quantity can be calculated.

Thank you for this comment. We have added text to clarify the main point of this example.

This example on page 34 is one we have encountered frequently in analyzing text describing patient smoking behavior. Here, we observe that physicians often use the units of “packs per year” which in general is a valid property. However, in the context of describing cumulative smoking history, the correct unit is “pack-years” defined as the number of packs smoked per day multiplied by the number of years smoked. Physicians and/or transcribers often confuse the meaning of these two properties. However, the readers of the reports typically understand that the convention is to report “pack-years” and understand that the mention of “packs-per-year” is often misused. (Pack-year descriptions are used to determine eligibility for lung cancer screening and as such is the important information the reporting physician is trying to communicate). A language understanding program for extracting patient smoking history thus needs to be able to correctly infer the intended meaning conveyed by the reporting physician despite the misuse of unit terminology.

Q-04: "Punctuation Mis-use" was mentioned as examples of language complexities in medical reports. However, does this bring problems to NLU system at all? Normally NLP systems would remove punctuations at a certain stage of data processing.

In general, there is a variety of cases where consideration of punctuation can improve the effectiveness of an NLP application. For example, certain punctuation is expected for indicating sentence and phrasal boundaries that are required inputs for tasks such as syntactic parsing. Without such expected punctuation, specific NLP routines such as sentence boundary detection (periods, question marks, exclamations, and ellipses), section boundary detection (e.g., colons), syntactic parsing, and symbolic expression interpretation (e.g., slashes, dashes, colons) can fail. Furthermore, punctuation marks themselves are often activations for triggering a particular type of analysis. 

Q-05: Example in "Temporal Ambiguity" is also not really ambiguity. It has to do with information not captured by the physician. 

The NLU problem is distinguished from general NLP in that it is tasked with inferring conveyed meaning, often with the support of real-world situational pragmatic knowledge. Thus, in addition to dealing with language specific issues, it also must be able to infer meaning of text in the light of a common knowledge understanding between writer and reader. For the example given on page 35, the information required by the reader managing the patient is whether the patient is a current smoker or not. (This is needed in order to determine if the patient needs smoking cessation counseling). This information needs to be rationalized based on the information of “no tobacco” and “smoked 3 packs per day x 17 years”. The correct temporal information that is inferred in this example is that the patient is not a current smoker, but previously smoked for 17 years.

Q-06: "Inferred information from practice guidelines" is not an examples of language complexities, but rather a knowledge complexity which will require periodic update of the knowledgebase or proper version control. I think it should be discussed separately as an essential part of system design.

As mentioned above, we include the requirement for integrating real world knowledge in order to infer the intent of the authoring physician (The NLU problem vs the NLP problem). The NLP problem only addresses language issues. The NLU problem must include the machinery to infer speaker intent, and as such we include rich pragmatic knowledge bases as part of the NLU architecture. The intent in this example (page 35) is to convey the severity of the patients smoking history in order to provide additional evidence for assessing lung cancer risk. 

Q-07: The author compared Cognitive Framework with deep learning based framework for NLU. While I do see the similarities between the two frameworks in terms of hierarchical structure, a more fair comparison would be with other previous content-based systems (or there can be other names used e.g., scenario-specific systems, systems enabling spatial and temporal constraints).

Thank you for this suggestion. There are a number of systems built for information retrieval, robotics, and question-answering applications that utilize situational context and pragmatic knowledge for optimizing end-user experience. While a comparison of the types of knowledge, their representation, and their methods for inferencing could provide valuable insight into the strengths and weaknesses of our design, we feel this comparison is beyond the scope of this paper. In this paper, we aim to emphasize the four foundational ideas of semantic memory, semantic composition, semantic activation, and hierarchical predictive coding. We provided conceptual illustrations of what each component involves, the rational for their inclusion, and an operational description of how they might be incorporated in a medical NLU implementation. We are writing, however, a follow-up paper on the implementation of our design for analyzing radiology reports and in that paper, we will provide a detailed comparison of related knowledge-based systems, as per your suggestion. 

Reviewer #4:

Q08: Overall, this concept paper reviews some ideas that might be useful for achieving a “deep and true” understanding of clinic texts, however, it lacks a detailed review for the applicable contexts and tasks. For a successful publication, I believe that this paper shall provide either theoretical or empirical contributions. I would strongly suggest that it shall include empirical support to justify the performance of the proposed framework. 

Thank you for this suggestion. In regard to the applicable contexts and tasks, we have included a recent reference to a comprehensive book chapter [Roberts 2022]. The chapter presents the traditional NLP tasks and various classes of Healthcare applications that have been research within the field over the past many decades. We also provide a summary list of common application areas on page 32 of the paper with several references.

Roberts K. Chapter 8 - Natural Language Processing, in Health Informatics, Practical Guide, 8th Edition, (William Hersh, ed.), ISBN 9781435787759, 2022

In regard to including empirical evidence, we initially drafted the paper showing a prototype of the system with a fully described semantic hierarchical knowledge based tuned for analysis of sentences describing tumoral masses and patient smoking habits. The analysis was based on 36K tumor description sentences and 10K smoking behavior sentences. Our internal review of the paper however led us to the conclusion of separately describing three viewpoints including the conceptual architectural design, the implementation methods, and application evaluation. In this paper, we chose to avoid the technical details of the implementation since it would distract from the theoretical discussions of the cognitive-inspired narrative of our design. Separate papers are planned for describing the implementation of a prototype implementation and for the evaluation of the approach in two clinical application areas (viz., characterization of radiographic tumoral descriptions and characterization of patient smoking behavior). In this paper, our goal was to make theoretical arguments for moving toward a cognitive architecture. We feel it is first necessary to explain to the medical informatics community the conceptual framework including its underlying foundational principles and their major strategic long-term advantages and disadvantages. 

Q09: Also, developing metrics for the proposed framework’s evaluation is important and human experts’ evaluations seem to be also essential in this matter. However, this paper only presents a general summary of evaluation required by medical NLU systems in P35. It is uncertain for the HSCM’s evaluation and so to its real effectiveness. 

Thank you for this question. The metrics associated with the architecture can be linked to the described arguments concerning its conceptual advantages and disadvantages. We have included a summarized of metrics from various perspectives in the Discussion section of the paper and summarized below: 

Human development effort point of view: Since the HSCM should parallel how humans model the semantics of words, objects, events, and topics, we envision much of its implementation will be manually defined by humans. The metrics to quantify this manual effort per application might include level of expertise, man-hours required for model development, and level of competence (e.g., errors, consistence) of HSCM authors in following guideline rules. 

From a knowledge engineering perspective: Evaluation metrics for building the HSCM knowledge base can be expressed in terms of traditional evaluation metrics used for ontologies and include expressiveness (representational adequacy), inferential adequacy (ability to infer new information), inferential efficiency, and acquisitional efficiency [Smith 2006].

Smith B. “From concepts to clinical reality: an essay on the benchmarking of biomedical terminologies,” Journal of Biomedical Informatics 39(3):288-298, 2006.

From a software engineering perspective: The HSCM defines a hierarchical graph that is based on semantic frames. It thus very much embodies the ideas of an object-oriented organization for classes and their associated methods. Thus, the important object-oriented features of inheritance, abstraction, encapsulation, modularity, recursion, and procedural triggers can be easily realized by the architecture. Metrics associated with object-oriented software systems are defined in [Chidamber 1994]. Performance metrics that can be defined from these features include time/effort to define new or edit existing HSCM nodes, and time/effort to debug definitional errors. Additional metrics related to the complexity of the HSCM graph include branching complexity, path complexity, data complexity and decisional complexity. These metrics become increasingly important as the breath of semantic constituents and their complexity rises.

Chidamber, S.R.; Kemerer, C.F. “A metrics suite for object oriented design,” IEEE Transactions on Software Engineering Volume 20, Issue 6, Jun 1994 Page(s):476 - 493

From a computation perspective: The general problem of NLU is difficult since it requires a mapping from all possible sentence inputs for a domain to all possible interpretations sanctioned by the software system. This results in a huge state space mapping. To tackle the “curse of dimensionality” issue, the system introduces structure in terms of hierarchical semantic composition. This allows the joint problem to be factored into a number of lower dimensional mappings. Computation time for the parser to search the HSCM for an optimal interpretation path is facilitated using a predictive coding algorithm. The search time savings is conceptually reduced from an exhaustive bottom-up search to a controlled hybrid search defined by only plausible hypotheses.

From an end-user clinical application perspective: See the evaluation metrics summarized on pages 35-36.

Q-10: The authors stated “… NLP/NLU of clinical reports is an important area in medical informatics … However, it is challenging to perform deep understanding of clinical notes by computers …” without giving a clear definition of “deep understanding”, what does it mean? Since this is the motivation of designing such framework, it is necessary to define the level of understanding the filed expects rather than simply saying “closer to human capabilities”. 

Thank you for pointing out this omission. We have included a more precise explanation for the goals of a deep understanding system as applied to medical text. Specifically, we have directed the reader to the section of the paper the details how performance requirements of an NLU system are defined with respect to the actionable task it is supporting, and the input text domain it intends to operate on.

The phrase deep understanding as common in NLU literature refers to systems that have the goal of human level understanding of a text. This requires situational awareness and generally implies the ability to infer meaning utilizing a wide range of language and domain knowledge. It is commonly distinguished between self-learning systems (e.g., deep learning architectures) that have been characterized as having “a mouth without a brain” and “statistical parrots” [Lederman 2022]. Bender has argued that deep learning systems lack the fundamental mechanisms for inferring speaker intention and thus the limitations of such architectures [Bender 2020].

Lederman A, Lederman R and Verspoor K. “Tasks as needs: reframing the paradigm of clinical natural language processing research for real-world decision support,” Journal of the American Medical Informatics Association, 29(10):1810-1917, 2022.

Bender EM and Koller A. Climbing toward NLU: on meaning, form, and understanding in the age of data,” Proceedings of the 58th Annual Meeting of the Association for Computational Linguistics, (Best Theme Paper), Pages 5185-5198, July 2020.

The problems that require natural language “understanding” (i.e., intended meaning) are those tied to patient outcomes. One important distinction within the medical informatics field is that the NLU system’s worth hinges on providing the necessary information to accomplish an “actionable” task. Medical NLU systems thus are not intended to be general broad coverage agents, but rather targeted agents that are tasked to understand text at sufficient levels of detail and content to correctly guide a clinical or research action. It is important to realize that these tasks can be high-stake and/or mission critical responsibilities, compared to an NLP system that may be searching for information stored in the web or in journal articles. For example, we have worked on applications for identifying patients who should be screened for lung cancer. Failure to identify such patients should not hinge on idiosyncratic language used in a medical report. Thus, ignoring tail distribution cases is often unacceptable. These difficult cases often require rich models of medical entities and procedures. Having a logical semantic substrate to integrate pragmatic and global situational models will improve an NLU ability to handle complex language phenomena (e.g., ellipses, coreference resolution, and linguistic paraphrasing). 

Q-11: Also, “true intended meaning” in such context lacks a clear definition. It will be good to give examples for some texts derived from the clinic reports and show both “wrong interpretations” VS “true intended meaning”.

Thank you for this suggestion. In the Introduction, we have now provided a note to the reader to see the Discussion Section, under Speaker-Listener Model of the paper which lists examples of various ambiguities that can lead to misunderstanding of clinical text. We describe various classes of ambiguity and potential situations for erroneous interpretations in this discussion. 

Q-12: The HSCM is designed for a sentence-level understanding, what is the reason for picking this level? Semantic understanding is important as stated in the Background section, so defining the semantics used in the HSCM model is essential for its success. Performing pilot studies to test the length of words will be helpful in this matter. The sentence-level model processing might not be an optimal selection, because the clinic texts/reports might be very context-sensitive, such as a good understanding only can be achieved via reading several paragraphs, thus, there might be some misunderstanding due to the selection of the unit.

Thank you for this important concern. The NLU problem involves synthesizing lower complexity representations to build increasingly complex and comprehensive meanings. Each structural level (words, phrases, clauses, sentences, paragraphs, reports, etc.) require different NLU methods to handle the particular semantic synthesis tasks presented within the text. We believe the sentence level is an appropriate central level of abstraction for building meaning for the following reasons. Firstly, there are strong regularities that can be exploited including syntax and topic focus. Secondly, the majority of logical semantic building blocks that are defined at the sentence level (concepts, objects, relations, events) cover a wide spectrum of what are needed to represent text over larger spans. Thirdly, methods for connecting thoughts across sentences (e.g., coreference resolution) involve making associations between logical representations at the sentence level. These methods involve semantic selectional constraints, object description plausibility, and situational expectations [McShane 2021]. The highest representations of the HSCM (e.g., e.g., discourse models, situational models) are intended to encode such knowledge required for resolving, for example, anaphoric ambiguities between nearby sentences.

Q-13: The introduction section lacks a statement of findings and a summary of contributions.

Thank you for this oversite. We have included the main contribution points of the paper in the Introduction.

Q-14: Medical NLP studies have lots of applications. This paper in general does not include an overview of the NLP/NLU systems, which is especially important for conducting a systematic knowledge of the NLP concepts, tasks, performance in the medical domain.

This intended audience for this paper is directed toward researchers within the medical NLP domain. We have briefly described some common tasks and methods used during the prior decades of the field. To limit the length of this paper, we have included a reference to a recent book chapter summarizing various topics applications of concern within the field [Roberts 2022] as well as a review paper on this topic [Gao 2022].

Roberts K. Chapter 8 - Natural Language Processing, in Health Informatics, Practical Guide, 8th Edition, (William Hersh, ed.), ISBN 9781435787759, 2022

Gao Y, Dligach D, Christensen L, Tesch S, Laffin R, Xu D, Miller T, Uzuner O, Churpek MM, and Afshar M. “A scoping review of publicly available language tasks in clinical natural language processing,” Journal of the American Medical Informatics Association 29(10:1797-1806, 2022.

Q-15: Definitions for performance metrics are missing. This paper proposes a framework (a concept paper) without conducting any experiments using “real clinic data”, so the argument “This structure enables a more efficient process of encoding sentence meaning, by facilitating a generative model.” is not supported from the overall study.

Please see the response to Q-09 in regard to performance metrics.

In making the statement “This structure enables a more efficient process of encoding sentence meaning by facilitating a generative model,” was made based on two theoretical grounds: composition (representational efficiency) and hierarchical predictive coding (processing efficiency). Imposing a compositional structure (i.e., factorization) is known to contribute significantly to reducing the dimensionality (i.e., computational complexity) of the parsing problem [see references by Geman 2002 and Yuille 2013]. It provides a framework for ‘part sharing’ which allows development to proceed in a piece-wise systematic way. (Of course, the up-front cost of such an approach is in curating this HSCM knowledge source for the target clinical application). As described in Yuille, this part sharing can lead to an enormous reduction in complexity. Predictive coding offers processing efficiency since only plausible hypotheses specified within the HSCM grammar need be tested. A combination of top-down (hypothesis formulation) and bottom up (hypothesis testing) processing conducted within a hierarchical predictive paradigm greatly reduces the search state space for a viable global sentence parse. Note that a purely bottom up (inverse problem) approach to semantic parsing is regarded as an ill-posed problem [Barton 1987]. The HSCM model provides the additional constraints in order to rule out a large number of interpretation possibilities.

S. Geman, D. Potter, and Z. Chi. Composition systems. Quarterly of Applied Mathematics, 60(4):707–736, 2002.

Yuille, A. L. and Mottaghi, R. (2013) ‘Complexity of representation and inference in compositional models with part sharing’, 1st International Conference on Learning Representations, ICLR 2013 - Conference Track Proceedings, pp. 1–13.

Barton EG, Berwick RC, and Ristad ES. Computational Complexity and Natural Language, Bradford Books, Cambridge, MA, MIT Press, 1987.

In terms of application, we have implemented a prototype system that performs a semantic parse of descriptions of tumoral masses. This work has been on-going for the past several years and is the main reason we have confidence in the ideas of this paper. We initially drafted a paper that described the tools and HSCM model for this application. However, we thought it best to separate the conceptual design issues from the detailed implementation and tools in order to concentrate on the issues from these separate perspectives. Our plan is to first publish this concept design paper with subsequent papers that will describe a prototype implementation of the design and applications of the system for tumoral mass descriptions and patient smoking behavior. 

Q-16: It is not clear that the innovative aspects of the HSCM, how does it different from other NLU methods that could effectively process clinic texts? As the authors stated, unstructured clinical texts need to be carefully handled to create a structured output, but why HSCM could potentially outperform other existing systems? Here is a lack of justification.

There are very few papers that describe the design architecture for incorporating the machinery required for deep understanding of clinical text. Most medical NLP systems today are based on either traditional pipeline architectures that include sentence boundary detection, tokenization, morphological analysis, syntactic analysis, semantic analysis, and discourse processing [Soan 2014, Roberts 2022] or deep learning architectures [Wu 2020]. As far as we know, there are no reports of cognitive-inspired architectures for medical text processing that incorporate the ideas of symbol grounding, semantic activation, semantic composition, and hierarchical predictive coding. Much of the Discussion section of the paper is directed towards the justification for these ideas specifically for handing the performance requirements within a clinical environment and as a flexible long-term design that can support group community development of applications. (See “Summary of Main Arguments in Favor of a Cognitive Framework for Medical NLU” in the Discussion Section of the paper). We have included the references below for readers to further compare NLP architectures currently in development.

Doan S, Conway M, Phuong TM, and Ohno-Machado L. “Natural language processing in biomedicine: a unified system architecture overview,” Clinical Bioinformatics, Chapter 16, R Trent (Ed.), Clinical Bioinformatics, Springer New York, New York, NY (2014), pp. 275-294

Roberts K. Chapter 8 - Natural Language Processing, in Health Informatics, Practical Guide, 8th Edition, (William Hersh, ed.), ISBN 9781435787759, 2022

Wu S, Roberts K, Datta S, Du J, Ji Z, Si Y, Soni S, Wang Q, Wei Q, Xiang Y, Zhao B, and Xu H. “Deep learning in clinical natural language processing: a methodological review,” Journal of the American Medical Informatics Association 27(3):457-470, 2020.

Q-17: Also, any potential implications and applications will be further developed based on the HSCM framework?

In the Conclusion section of the paper, we have stated that we will be reporting on the application of the system for analyzing tumoral mass descriptions in radiology reports. We will also be describing an application for analyzing patient smoking behavior.

Q-18: Figure 4, describing the model architecture with various level managers, has many abbreviations, however, there is a lack of explanation for their meanings.

Thank you for this comment. The many abbreviations shown in Figure 4 are the frame labels we assigned to the various semantic nodes within the HSCM. We have included descriptions of the frame labels within the caption of Figure 4.

• pName.size – property name size

• Ont.Cnpt – Ontologic concept

• Num.real – real number

• PName – property name

• PValue – property value

• Ont.E-Frame – Ontologic entity frame

• POS.art.indef – Part-of-Speech, article, indefinite

• Locative.prep – locative preposition

Q-19: There is an underline for the word “understanding” on P3 line 55, which looks weird.

We have removed the underline.

Q-20: Please insert figures to the corresponding places of the manuscript. The authors put all the figures at the end of the manuscript which causes large inconvenience for reading.

We apologize for the inconvenience. However, this format is the style stated in the PLOS-ONE author’s manual.

---

## [Editor Report · Decision Letter 2]

27 Feb 2023

Design considerations for a hierarchical semantic compositional framework for medical natural language understanding

PONE-D-20-39301R2

Dear Dr. Taira,

We’re pleased to inform you that your manuscript has been judged scientifically suitable for publication and will be formally accepted for publication once it meets all outstanding technical requirements.

Kind regards,

Ilya Safro, Ph.D.

Academic Editor

PLOS ONE

---

## [Editor Report · Acceptance letter]

6 Mar 2023

PONE-D-20-39301R2 

Design considerations for a hierarchical semantic compositional framework for medical natural language understanding 

Dear Dr. Taira:

I'm pleased to inform you that your manuscript has been deemed suitable for publication in PLOS ONE. Congratulations! Your manuscript is now with our production department. 

Kind regards, 

on behalf of

Dr. Ilya Safro 

Academic Editor

PLOS ONE